# Position: Theory of Mind Benchmarks are Broken for Large Language Models

**Matthew Riemer** [1 2 3]  **Zahra Ashktorab** [1]  **Djallel Bouneffouf** [1]  **Payel Das** [1]  **Miao Liu** [1]  **Justin D. Weisz** [1]
**Murray Campbell** [1]

## Abstract

Our paper argues that the majority of theory of mind benchmarks are broken because of their inability to directly test how large language models (LLMs) adapt to new partners. This problem stems from the fact that theory of mind benchmarks for LLMs are overwhelmingly inspired by the methods used to test theory of mind in humans and fall victim to a fallacy of attributing human-like qualities to AI agents. We expect that humans will engage in a consistent reasoning process across various questions about a situation, but this is known to not be the case for current LLMs. Most theory of mind benchmarks only measure what we call *literal theory of mind*: the ability to predict the behavior of others. However, this type of metric is only informative when agents exhibit self-consistent reasoning. Thus, we introduce the concept of *functional theory of mind*: the ability to adapt to agents in-context following a rational response to their behavior. We find that many open source LLMs are capable of displaying strong literal theory of mind capabilities, but seem to struggle with functional theory of mind – even with exceedingly simple partner policies. Simply put, strong literal theory of mind performance does not necessarily imply strong functional theory of mind performance or vice versa. Achieving functional theory of mind, particularly over long interaction horizons with a partner, is a significant challenge deserving a prominent role in any meaningful LLM theory of mind evaluation.

## 1. Introduction

Many recent papers that evaluate theory of mind in LLMs have been inspired by how humans are evaluated for theory

---
[1] IBM Research [2] Mila [3] Université de Montréal. Correspondence to: Matthew Riemer <mdriemer@us.ibm.com>.

*Proceedings of the 42nd International Conference on Machine Learning*, Vancouver, Canada. PMLR 267, 2025. Copyright 2025 by the author(s).

of mind (Bubeck et al., 2023; Kosinski, 2023; Ma et al., 2023; Street et al., 2024; Strachan et al., 2024). Although this course of action seems logical, it is important to recognize that AI has a tendency to over optimize for its training objectives in a manner that is quite alien to the way the human brain works. For example, Bubeck et al. (2023) tout the performance of LLMs on human theory of mind tests, but they note a noticeable lack of "process consistency" in LLM explanations: LLMs can come up with compelling explanations for what they do that have very little to do with their actual reasoning process. We must therefore proceed with caution about our conclusions when we evaluate LLMs in ways that do not directly align with our objectives. When we evaluate humans, we typically focus on what we call in this paper *literal theory of mind*, which is their ability to predict the behavior of other agents. As noted by Ma et al. (2023), this concept encompasses various abstractions including actions, intentions, beliefs, percepts, desires, knowledge, and emotions. However, with humans we take for granted that the ability to predict the behavior of other agents will be consistently applied within their own reasoning process when determining their own behavior. The main insight of our paper is that this form of process consistency cannot be taken for granted when evaluating LLMs.

Our position is: **Current LLM theory of mind benchmarks are broken because they only measure literal theory of mind (Definition 2.1) rather than functional theory of mind (Definition 2.2), which misses a key aspect of evaluating an agent's ability to choose appropriate actions based on the behavior of another agent.** In Section 4 we demonstrate simple, concrete cases where prominent open source LLMs display strong literal theory of mind performance that is illusory in nature because it does not also lead to strong functional theory of mind performance. Our results serve as a case study to highlight the misleading nature of most LLM theory of mind benchmarks even when we do find that agents achieve impressive performance.

Recently, generative AI, particularly in the form of LLM assistants, has been deployed as a tool for a growing variety of real-world use cases where the LLM must interact with a diverse set of people performing a diverse set of tasks. As typically deployed today, these LLMs are only interacting with users at inference time due to the significant computa-

tional cost of continuing to train these large models in the context of individual users. As a result, LLMs must learn to adapt their behavior to users in-context based on recorded interaction histories. Indeed, a similar scenario of growing importance involves LLMs interacting with other AI agents through an agentic framework (Shavit et al., 2023). Our work is inspired by the work of Akata et al. (2023) to leverage canonical repeated games from behavioral game theory as a way to asses the adaptation ability of LLMs across a full spectrum of incentive structures. Unfortunately, we find significant deficiencies in the ability of prominent open source LLMs to adapt to new partners in-context, providing a sobering analysis of the ability of LLMs to reliably adapt without continual training of the model weights themselves.

The term "theory of mind" is used in a variety of different ways across different bodies of literature. We use the term as an idiomatic description of behavior, although it can also be interpreted more literally to mean there is an actual artifact called a "theory of mind" that depicts an explicit description or model of an agent's mental state. *Literal theory of mind* can then alternatively be considered as theory of mind *prediction*, and it is possible to achieve optimal precision without forming an explicit description of an agent's mental state. For example, a model may be conditioning on a variable that is perfectly correlated with the variables another agent is actually considering. The literature on theory of mind in machine learning is generally unconcerned about this distinction. If this prediction ability is achieved, it is not of practical consequence if there actually is a theory of mind model. Likewise, *functional theory of mind* can alternatively be considered to be theory of mind *reasoning*. Our assertion is that *functional theory of mind* is what the AI research community should be focused on creating and evaluating. Notice that this goal requires the ability to adapt to other agents, but does not necessarily require that the agent has an explicit theory of mind model. If we are describing a capability, it is important to focus on the problem it solves, which must be disentangled from the solution method. To define the capability of theory of mind in such a way that an explicit theory of mind model is needed would only make sense if there were kinds of behaviors expressible with model-based approaches that are not expressible with model-free approaches. However, we know that these algorithm classes have the same expressiveness – at least in the case of reinforcement learning (Sutton & Barto, 2018).

The position of our paper was inspired by an unexpected result in our early experiments. We considered the incredibly simple scenario of an LLM agent playing the classic game Rock, Paper, Scissors against an agent that always plays the action "Rock" for 100 consecutive rounds. The optimal course of action is for the LLM to respond with the counter to this action "Paper" as much as possible. It would be reasonable for an agent to take a few exploratory

actions, but there would be little excuse for not winning a high percentage of the 100 rounds. What we found is that a vanilla application of common open source LLMs resulted in a policy that chose each action "Rock," "Paper," and "Scissor" roughly evenly. This result is interesting because it is actually the famous Nash equilibrium solution for this game. However, this solution optimizes the worst case return across any possible opponent. To act in this way against an opponent that always plays "Rock" for 100 consecutive rounds actually demonstrates a profound lack of theory of mind. But it should be emphasized that this LLM agent does actually display a high-level of *literal* theory of mind. Indeed, it only takes so many rounds of seeing the other agent take the "Rock" action before the LLM is able to predict it will keep doing so. The problem is that this prediction has nothing to do with its own chosen course of behavior. We believe this is an important insight for the AI community to grapple with when measuring theory of mind in agents, and we formalize the concept of *functional theory of mind* to accurately describe this often lacking capability.

## 2. Two Types of Theory of Mind Between Multi-Agent LLM Policies

**The Environment.** The interaction process between multiple agents is often formalized as a stochastic game (Shapley, 1953) or rather a Markov game in the fully observable setting (Littman, 1994). For generality in this work, we consider the setting of partially observable stochastic games (POSGs) (Kuhn, 1953) that generalize POMDPs (Kaelbling et al., 1998) and MDPs (Puterman, 1994) to the multi-agent and decentralized setting. A POSG can be defined as the tuple $< \mathcal{I}, \mathcal{S}, \mathcal{A}, \mathcal{T}, \mathcal{R}, \mathcal{X}, O, T >$. Here $\mathcal{I}$ is a finite set of agents, $\mathcal{S}$ is a finite set of global states, and $\mathcal{A} = \times_i \mathcal{A}^i$ is the set of joint actions across agents $i \in \mathcal{I}$. $\mathcal{T} : \mathcal{S} \times \mathcal{A} \mapsto \mathcal{S}$ is the state transition function based on the joint actions across agents, and $\mathcal{R} = \times_{i \in \mathcal{I}} \mathcal{R}^i$ is the joint reward function with reward function $\mathcal{R}^i : \mathcal{S} \times \mathcal{A} \mapsto \mathbb{R}$ for each agent $i \in \mathcal{I}$. $\mathcal{X} = \times_i \mathcal{X}^i$ is the joint set of observations with $\mathcal{X}^i$ denoting a finite set of observations for each agent. $O : \mathcal{S} \mapsto \mathcal{X}$ is the function that produces agent observations based on the state. Finally, $T$ is the horizon of interactions before termination.

**Agent Interaction.** We can now consider environment interaction at each step from the perspective of a given focal agent $i \in \mathcal{I}$ where we will use $-i := \mathcal{I} \setminus i$ to denote set of all other agents. At step $t$ agent $i$ takes action $a_t^i$ and the other agents take a joint action $\boldsymbol{a}_t^{-i}$ yielding a transition from $s_t$ to $s_{t+1}$ with probability $\mathcal{T}(s_{t+1}|s_t, \boldsymbol{a}_t)$ where $\boldsymbol{a}_t = \{a_t^i, \boldsymbol{a}_t^{-i}\}$. The agent then receives its own observation of this state $x_{t+1}^i$ and reward $r_t^i$. In the environments we consider in this paper it is assumed that $x_{t+1}^i$ also contains information about the actions of the other agents at the previous step $\boldsymbol{a}_t^{-i}$. However, this need not always be the case. In such a

POMDP, each agent's policy $a_t^i \sim \pi^i(h_t)$ generates its action stochastically based on its own interaction history $h_t^i$ where $h_t^i := \{x_1^i, a_1^i, r_1^i, ..., x_t^i\}$. Note that policies being defined in this way subsumes the common case where part of the history is discarded for computational or memory efficiency.

**LLM History Representations.** As we are interested in evaluating LLMs for this problem it is additionally assumed that this history representation $h_t^i$ must be encoded in text (i.e. LLM tokens). This implies that there must be a function that we have direct access to converting the observations $x_t^i$, actions $a_t^i$, and rewards $r_t^i$ to the form of LLM token representations. Moreover, it is important to note that we are interested in learning generalist policies across tasks with LLMs. As a result, following the protocol of Akata et al. (2023), it can also be useful to include any known information about the environment tuple $< \mathcal{I}, \mathcal{S}, \mathcal{A}, \mathcal{T}, \mathcal{R}, \mathcal{X}, O, T >$ to the agent to promote rapid adaptation to new tasks.

## 2.1. Disentangling Literal Theory of Mind from Functional Theory of Mind Evaluation

The goal of any individual agent $i$ interacting in an environment for $T$ steps is to learn a policy $\pi^{i*}$ that maximizes its expected reward given the policies of the other agents in the environment $\boldsymbol{\pi}^{-i}$ starting from state $s_1$:

$$\pi^{i*} = \arg\max_{\pi^i} \mathbb{E}\left[\sum_{t=1}^{T} r_t^i | s_1, \boldsymbol{\pi}^{-i}\right].$$

Because the best course of action $\pi^{i*}$ is critically dependent on the policies of the other agents $\boldsymbol{\pi}^{-i}$, it is common for agents to directly learn a model of the other agent's policies $\hat{\boldsymbol{\pi}}^{-i}$ in decentralized settings. This is analogous to model-based RL (Sutton & Barto, 2018; Brafman & Tennenholtz, 2002; Schrittwieser et al., 2020) and is useful for stabilizing learning (Lowe et al., 2017). However, it is worth noting that it is not a strict requirement for representing a policy $\pi^{i*}$ that is optimal in functionality (Sutton, 1991). That said, it is tempting to consider when using LLMs as it can be generated by simply prompting the model with a token representation of the interaction history $h_t^j$ for $j \in -i$. We can then model performance of any particular approximate *literal theory of mind* model using Definition 2.1.

---

**Definition 2.1** (*T*-Step Literal Theory of Mind Loss)**.** The loss from start state $s_1$ with respect to a joint policy $\boldsymbol{\pi}^{-i}$ that generates $T$ actions $a_1^{-i}, ..., a_T^{-i}$ of its approximation $\hat{\boldsymbol{\pi}}^{-i}$ that generates $T$ actions $\hat{a}_1^{-i}, ..., \hat{a}_T^{-i}$ is:

$$\mathcal{L}_{\text{Literal}}(s_1, \hat{\boldsymbol{\pi}}^{-i}, \boldsymbol{\pi}^{-i}, T) =$$
$$\mathbb{D}(\phi(a_1^{-i}, ..., a_T^{-i}), \phi(\hat{a}_1^{-i}, ..., \hat{a}_T^{-i}))$$

where $\mathbb{D}$ is some distance function and $\phi$ is some abstraction mapping function over actions.

---

We phrase Definition 2.1 in terms of an abstract distance

function $\mathbb{D}$ and abstraction mapping function $\phi$ to keep the definition as broad as possible and encompass as much as we can of the existing literature on theory of mind. Primitive actions can definitely be directly considered (as we do in this paper). In this case, $\phi$ is the identity mapping and $\mathbb{D}$ is set to a percent error metric. $\phi$ can also be straightforwardly set to various temporal abstractions of actions given sufficient history length $T$. Moreover, latent features that impact the behavior of agents such as intentions, beliefs, percepts, desires, knowledge, and emotions can also be considered as alternative for $\phi$ – as an inverse mapping must exist. However, the agent's actions alone may not provide enough information, for example, to fully determine an agent's emotional state, we can only judge the theory of mind performance of an agent in terms of the information provided. This does not serve as meaningful limitation of this definition because Bayesian reasoning is generally the desired outcome in the presence of lack of information.

One obvious issue with Definition 2.1 is that it is a function of a theory of mind model $\hat{\boldsymbol{\pi}}^{-i}$, which is not even necessarily needed to express the policy of an agent $\pi^i$. For cases where the value of $\hat{\boldsymbol{\pi}}^{-i}$ is decoupled from the reasoning involved in $\pi^i$, it is of particular importance to consider a functional metric of the degree to which the policy $\pi^i$ is catered to the particular other agent policies $\boldsymbol{\pi}^{-i}$. We can define this metric in terms of the $T$-step regret incurred by $\pi^i$.

---

**Definition 2.2** (T-Step Functional Theory of Mind Regret)**.** The $T$ step regret from start state $s_1$ of policy $\pi^i$ that receives individual rewards $r_1^i, ..., r_T^i$ when playing with policy $\boldsymbol{\pi}^{-i}$ in comparison to the optimal policy $\pi^{i*}$ that plays the optimal $T$-step response to policy $\boldsymbol{\pi}^{-i}$ and receives rewards $r_1^{i*}, ..., r_T^{i*}$:

$$\Delta_{\text{Functional}}(s_1, \pi^i, \pi^{-i}, T) = \sum_{t=1}^{T} (r_t^{i*} - r_t^i).$$

---

Definition 2.2[1] provides us with a functional metric for measuring theory of mind in the presence of other agents parameterized by $\boldsymbol{\pi}^{-i}$. However, it is still worth considering when the conclusions from this metric will be much different than conclusions from a literal theory of mind metric following Definition 2.1. To do this we must define a new policy $\pi_{\text{ToM}}^i$ that is directly based on the literal theory of mind model prediction $\hat{\boldsymbol{\pi}}^{-i}$ such that behavior is chosen to be a rational maximization of the action-value function. In our experiments we simulate such a policy utilizing ground truth knowledge of the payoff structure and use $\Delta_{\text{ToM}}(s_1, \hat{\boldsymbol{\pi}}^{-i}, \pi^{-i}, T)$ to denote the regret of such a policy. $\Delta_{\text{ToM}}$ thus represents the best regret that can be possibly achieved by faithfully following the predictions of our theory of mind model under

---

[1]The regret $\Delta$ can also be considered a form of loss for consistency with Definition 2.1. We chose this notation, rather than $\mathcal{L}$, to stay consistent with the common notation for each in the literature.

the assumption that the predictions are correct.

**Interesting Tasks for Theory of Mind.** Definition 2.2 also provides us with a natural way to characterize how interesting a task is for showcasing theory of mind. Specifically, the task definition must include distributions over the joint policies of the other agents $p(\pi^{-i})$ and start states $p(s_1)$. We can then consider an expectation over two joint policies $\pi'^{-i}, \pi''^{-i} \sim p(\pi^{-i})$ drawn from the distribution and define $\pi'^*$ as the optimal response to $\pi'^{-i}$. The metric would then be $\mathbb{E}_{s \sim p(s_1), \pi'^{-i}, \pi''^{-i} \sim p(\pi^{-i})}[\Delta_{\text{Functional}}(s, \pi'^*, \pi''^{-i}, T)/T]$. This metric requires that the task itself highlights the need to adapt to the specific policies of the other agents. For example, notice that for Rock, Paper, Scissors against arbitrary one action policies, as described in Section 1, this metric evaluates to a significant 1.0 expected regret per step when the reward for wins is $+1$, 0 for ties, and $-1$ for losses.

**Metrics for Our Experiments.** In our experiments, we aim to get a holistic view of both the literal theory of mind and functional theory of mind performance of each LLM and prompting strategy. We report the accuracy of the literal theory of mind predictions with respect to individual actions as ToM %. We also report the regret per step functionally achieved by each policy as $\Delta_{\text{Functional}}/T$. Finally, to get a clearer picture of the difference between the literal theory of mind performance and functional theory of mind performance, we report $\Delta_{\text{ToM}}/T$ the regret per step of the rational policy based on the literal theory of mind model.

**Accumulated Reward vs. Regret.** In the experiments of our paper, the games are simple enough that it is easy to compute the reward rate of the optimal policy. However, in most cases with complex interactions the optimal policy is unknown or even unknowable. In these cases, the accumulated reward is commonly used in place of regret. This is also a valid metric of functional theory of mind as it only differs from the regret by a constant factor based on the accumulated reward of the optimal policy. The advantage of using regret, when possible, is to directly keep track of the relative optimality.

### 2.2. Initial Results

We begin by conducting experiments on the Rock, Paper, Scissor domain as discussed in Section 1. Our goal here is to conduct an experiment that is as simple as possible so that we can clearly disentangle *literal theory of mind* from *functional theory of mind* without additional conflating factors. LLAMA-2 had previously been found to achieve strong performance on repeated matrix games (Lorè & Heydari, 2023), so we considered initial experiments with the full LLAMA-2 family (Touvron et al., 2023) as well as the competitive Falcon 40B (Almazrouei et al., 2023) and Mixtral models (Jiang et al., 2024). Our results are provided in Table 1. Our prompting strategy for the LLM policy $\pi^i$ follows what was established for repeated matrix games by Akata

et al. (2023) (see Figure 1). Our prompting strategy for the theory of mind model is detailed in Figure 2. The horizon of interaction is $T = 100$ steps against a randomly chosen fixed single action policy i.e. always Rock, always Paper, or always Scissors. Note that this interaction horizon is 10x longer than prior work (Akata et al., 2023) as we want to make sure behavior converges. 95% confidence intervals of the mean estimate across opponent policies are provided. See Appendix C for further details.

> **Functional Theory of Mind Gap**
>
> Throughout Table 1 we see a massive gap between the regret based on a rational response to the theory of mind model $\Delta_{\text{ToM}}$ with accuracy ToM % and the functional regret achieved in practice $\Delta_{\text{Functional}}$.

We also implemented a simple tabular model that performs efficient exploration following the classic RMax algorithm (Brafman & Tennenholtz, 2002) for near optimal worst case sample efficiency driven by optimistic Q-Learning (Watkins, 1989) with the actions at the last time step treated as the state representation. The tabular theory of mind model is based on a simple frequency count based prediction using the same state representation. It is clear that no LLM model gets even close to the performance of the tabular model in terms of functional theory of mind performance – although some come close in terms of their literal theory of mind performance. We find that Mixtral outperforms the LLAMA-2 family and we will thus subsequently include experiments with LLAMA-3 (which performs better). However, it is also interesting to see the trends within the LLAMA-2 family of models. Bigger models seem to get better performance and models with instruction tuning (labeled as "Chat" models) tend to get worse literal theory of mind performance, but better functional theory of mind performance. This is an intuitive result as instruction tuning further prioritizes interactive decision making and takes the model further from its original language modeling objective that is key to predicting what will come next in a sequence of interactions.

## 3. Overview of Existing Benchmarks

**Frameworks for Machine Theory of Mind.** In the field of multi-agent reinforcement learning, theory of mind is generally interpreted as the ability to directly predict the behavior (i.e. the actions) that another agent will take (Lowe et al., 2017; Rabinowitz et al., 2018). However, there can be a number of different ways to abstractly represent this behavior (Ma et al., 2023). For example, abstraction can be applied in the temporal dimension representing behavior as a composition of hierarchical skills (Sutton et al., 1999; Bacon et al., 2017; Riemer et al., 2018b; 2020; Allen et al., 2020; Abdulhai et al., 2021) or goals (Schaul et al., 2015; Andrychowicz et al., 2017; Yang et al., 2018; Kim

| LLM Model | $\Delta_{\text{Functional}}/T$ | $\Delta_{\text{ToM}}/T$ | ToM % |
|---|---|---|---|
| Tabular | **0.083** $\pm$ 0.004 | **0.039** $\pm$ 0.006 | **97.4** $\pm$ 0.4 |
| LLAMA-2 70B Chat | 0.857 $\pm$ 0.142 | 0.119 $\pm$ 0.006 | 92.1 $\pm$ 0.4 |
| LLAMA-2 70B | 0.971 $\pm$ 0.067 | 0.048 $\pm$ 0.003 | 96.8 $\pm$ 0.2 |
| LLAMA-2 13B Chat | 0.891 $\pm$ 0.173 | 0.095 $\pm$ 0.006 | 93.7 $\pm$ 0.4 |
| LLAMA-2 13B | 1.015 $\pm$ 0.044 | 0.049 $\pm$ 0.003 | 96.7 $\pm$ 0.2 |
| LLAMA-2 7B Chat | 0.904 $\pm$ 0.151 | 0.085 $\pm$ 0.004 | 94.3 $\pm$ 0.3 |
| LLAMA-2 7B | 0.972 $\pm$ 0.039 | 0.066 $\pm$ 0.004 | 95.6 $\pm$ 0.3 |
| Falcon 40B | 0.973 $\pm$ 0.040 | 0.056 $\pm$ 0.005 | 96.2 $\pm$ 0.3 |
| Mixtral 8x7B Instruct v1 | 0.542 $\pm$ 0.070 | 0.050 $\pm$ 0.005 | 96.7 $\pm$ 0.3 |

Table 1: Initial results for Rock, Paper, Scissors against simple single action policies.

et al., 2019a;b). Abstraction can also be applied at the agent level in order to represent the abstract actions of groups (Memarian et al., 2022; Touzel et al., 2024). Moreover, as showcased by the games Hanabi (Bard et al., 2020; Nekoei et al., 2021; Malloy et al., 2021a; Nekoei et al., 2023) and Poker (Ganzfried & Sandholm, 2015; Brown & Sandholm, 2019), theory of mind can be directly related to inferring recursive beliefs about unobserved information or knowledge (Moreno et al., 2021). We consider the efficacy of predictions of all of these representations of behavior as examples of measuring literal theory of mind (Definition 2.1). However, it has been recently recognized in the human-computer interaction (HCI) literature that we must go even further to achieve *mutual theory of mind* for better collaboration with humans (Wang & Goel, 2022; Zhang et al., 2024). In this framework, the mutual shaping of mental representations as well as the functional goal of improved performance on tasks are emphasized. We are inspired by this aspirational goal in our paper, leading us to define a functional measurement of theory of mind performance (Definition 2.2).

**Measuring LLM Theory of Mind.** While a number of recent studies have touted superior human-level LLM theory of mind capabilities, to the best of our knowledge these tasks have always center around passive question answering (Bubeck et al., 2023; Strachan et al., 2024; Kosinski, 2023; Street et al., 2024) as typified by the classic Sally-Anne false-belief test (Baron-Cohen et al., 1985). LLMs have also been successfully employed to simulate a diversity of personas (Park et al., 2024; Ha et al., 2024), trust behaviors (Xie et al., 2024), or even for macro-economic simulations (Li et al., 2023). However, these tasks all lack interactivity and only reflect strong performance at literal theory of mind (Definition 2.1). On the other hand, Kim et al. (2023) found that LLMs perform poorly when subjected to multiple question types that demand the same consistent underlying reasoning. This finding is in the same spirit and complimentary to the finding of our paper as well as the finding that LLMs lack process consistency (Bubeck et al., 2023). Moreover, LLMs have been found to fail at important interactive applications such as adaptive eduction for users of diverse age or education levels (Rooein et al., 2023) and providing coding assistance

for beginner programmers (Nguyen et al., 2024).

**Behavioral Game Theory with LLMs.** Our analysis of functional theory of mind in LLMs closely follows work considering LLMs in the context of behavioral game theory. However, this work has not featured contextualization of literal theory of mind performance and is not included as an aspect of popular benchmarks focusing on measuring literal theory of mind. Our paper builds off the work of Akata et al. (2023) who considered LLMs playing the Prisoner's Dilemma and Battle of Sexes. However, we focus on performance playing with simple policies rather than playing with other LLMs. LLMs will generally tend to generate in-distribution interaction histories when they play themselves, so we also find it more interesting to focus on very simple relatively out of distribution partner policies. For the Battle of Sexes game, Akata et al. (2023) found that GPT-4 only could adapt the canonical human-like alternating policy when using a special form of social cognition prompting that considers its own predictions of the actions of others (similar to "perspective taking" prompting (Xu et al., 2024)). We also tried this approach in our setting and did not find it to be an adequate fix the issues that LLMs experience. The ability for LLMs to play Rock, Paper, Scissors was previously considered by Fan et al. (2024) and the ability for LLMs to play the Prisoner's Dilemma was previously considered by Lorè & Heydari (2023), but not in the context of theory of mind.

**What We are Advocating For.** Our main insight is that benchmarks need an interactive component where the theory of mind reasoning performance of LLMs can be assessed separately from predictions about literal theory of mind. Games such as Codenames (Bills et al., 2025), Hanabi (Bard et al., 2020), Taboo (Ashktorab et al., 2020), and Wavelength (Morrison et al., 2025) are good examples of games that would score highly in terms of the relevance metric we proposed in Section 2. In Appendix A, we describe how literal and functional theory of mind could be measured in each of these games. In our experiments, we focus on simple matrix games in which it is even easier to disentangle these concepts and control for conflating factors.

# 4. Further Analysis: Difficulties in Achieving Functional Theory of Mind

In this section, we attempt to gain a better understanding of the surprising result in Section 2.2. Rocks, Paper, Scissors (RPS) is a canonical competitive game – as can be seen by the payoff table in Figure 5. We also wanted to evaluate our models on a canonical cooperative game, for which we chose the Iterated Battle of Sexes (IBS) following Akata et al. (2023) with payoffs detailed in Figure 6. Once again following Akata et al. (2023), we also evaluate our models on the famous Iterated Prisoner's Dilemma (IPD) mixed incentive game with payoffs detailed in Figure 7. We opted to test each model on the full spectrum of incentive structures to rule out explanations for performance lacking in competitive games such as an intrinsic altruism bias in LLMs (as has been previously suggested by Leng & Yuan (2023)).

## 4.1. Different Prompting Strategies

We consider three high-level classes of prompting strategies in order to provide a comprehensive evaluation.

**Generic Strategies:** We will henceforth call the prompting strategy used in Section 2.2 *LM Prompting* (see Figure 1) where the probability of each action in the action space is explicitly drawn from the model to generate the next action via next token prediction. See Figure 2 for the literal theory of mind version of the prompt. In *QA Prompting* (Figures 4 and 5), decision making is posed as a question answering problem where the LLM keeps generating stochastic outputs until they are in the desired format and action space vocabulary. This is implemented as a step along the way to Chain of Thought (CoT) reasoning (Wei et al., 2022) that we refer to as *CoT Prompting* (Figures 6 and 7) in which the LLM must generate a reasoning process and its own answer based on that reasoning process in the correct format.

**Strategies for In-Context RL:** We also wanted to consider prompting strategies in the literature specifically targeted at adapting LLMs in context to maximize reward. Fish et al. (2024) introduced *Plans + Insights Prompting* as a way to promote coordination between LLM agents based on GPT-4 by having the agents also generate text related to plans and insights and not just actions at each step while feeding in these generated plans and insights an input at the next step (see Figures 8 and 9). The idea is to be helpful in promoting consistent strategies across LLM calls. We also consider *Reflexion Prompting* (Shinn et al., 2024), which is a method for performing so-called verbal RL in-context by keeping a memory of reflections updated each time feedback is received from the environment. We provide example prompts in Figures 10, 11, and 12 and consider memory sizes between 1 and 3 (we always report the best value).

**Strategies Explicitly Considering Theory of Mind:** Fi-

nally, we consider *Social Prompting* as proposed for IBS by Akata et al. (2023) where the LLM first generates a prediction of the other agent's action and then conditions its reasoning based on that action. This approach is detailed in Figure 18 with the variant using QA prompting for the action prediction (Figure 5) being called *Social QA Prompting* and the one using LM Prompting for the action prediction (Figure 3) being called *Social LM Prompting*.

Our results applying these strategies to Mixtral 8x7B Instruct v1 are included in Table 2. We also provide comprehensive results in Appendix C.2 including for the LLAMA-2 70B Chat and LLAMA-3 70B Instruct models (Table 8) and Mistral Large 2 (Table 9). We find that LM Prompting always achieves the best literal theory of mind performance. Adding the QA format seems to make literal theory of mind a lot worse by taking the input distribution further from the LLM task and closer to the kind of processing needed for actual decision making. The gap between functional theory of mind and literal theory of mind performance is predictably smaller in this case. CoT Prompting has an inconsistent effect on literal theory of mind and always appears to make the functional theory of mind performance better than vanilla QA Prompting. Reflexion and Plans+Insights also seem to improve on QA prompting in terms of functional theory of mind performance, although we find that they do not quite match the performance of generic CoT prompting.

> **Gap Remains with Literal Theory of Mind Inputs**
>
> In Table 2 even when the literal theory of mind predictions are given as input, the LLM does not perform effective rational reasoning with this input.

Perhaps the most interesting case is Social Prompting, which seems consistently helpful regardless of the method of literal theory of mind prompting. However, a large gap between functional theory of mind and literal theory of mind remains even with Social Prompting. This is counterintuitive because the literal theory of mind model is directly provided as an input. It appears that the LLM still does not generate rational responses to these predicted actions. Note that the discrepancy in the LM Prompting ToM % comes from the need to prompt before agent's action is generated for Social Prompting (Figure 3) while prompting after the action is generated (Figure 2) is more effective when possible.

## 4.2. Reasoning Over Long Contexts

Due to the surprising result of Social Prompting still experiencing a significant functional theory of mind gap, we wanted to understand more about the difficulty these LLMs may experience when reasoning over a long context. We now consider the LLAMA-3 70B Instruct model (Dubey et al., 2024) for its superior performance over long context

| Prompting | Game | $\Delta_{\text{Functional}}/T$ | $\Delta_{\text{ToM}}/T$ | ToM % |
|---|---|---|---|---|
| LM | RPS | $0.542 \pm 0.070$ | $0.050 \pm 0.005$ | $96.7 \pm 0.3$ |
| QA | RPS | $0.881 \pm 0.117$ | $0.606 \pm 0.043$ | $59.6 \pm 2.9$ |
| CoT | RPS | $0.648 \pm 0.042$ | $0.998 \pm 0.020$ | $33.5 \pm 1.4$ |
| Plans + Insights | RPS | $0.705 \pm 0.058$ | $0.443 \pm 0.041$ | $70.5 \pm 2.7$ |
| Reflexion | RPS | $0.759 \pm 0.033$ | $0.530 \pm 0.042$ | $64.7 \pm 2.8$ |
| Social QA | RPS | $0.676 \pm 0.058$ | $0.619 \pm 0.045$ | $58.7 \pm 3.0$ |
| Social LM | RPS | $0.643 \pm 0.058$ | $0.119 \pm 0.038$ | $92.0 \pm 2.5$ |
| LM | IBS | $2.055 \pm 0.391$ | $0.216 \pm 0.026$ | $97.3 \pm 0.4$ |
| QA | IBS | $2.518 \pm 0.422$ | $1.818 \pm 0.259$ | $75.9 \pm 4.0$ |
| CoT | IBS | $2.169 \pm 0.107$ | $1.659 \pm 0.095$ | $79.8 \pm 1.4$ |
| Plans + Insights | IBS | $2.806 \pm 0.279$ | $1.800 \pm 0.274$ | $75.5 \pm 4.1$ |
| Reflexion | IBS | $2.590 \pm 0.179$ | $1.726 \pm 0.270$ | $77.9 \pm 4.2$ |
| Social QA | IBS | $2.557 \pm 0.451$ | $2.190 \pm 0.383$ | $70.3 \pm 5.7$ |
| Social LM | IBS | $2.082 \pm 0.329$ | $0.182 \pm 0.057$ | $97.4 \pm 0.8$ |
| LM | IPD | $0.949 \pm 0.142$ | $0.098 \pm 0.029$ | $97.8 \pm 0.5$ |
| QA | IPD | $2.365 \pm 0.179$ | $1.342 \pm 0.177$ | $68.8 \pm 2.7$ |
| CoT | IPD | $0.955 \pm 0.115$ | $1.105 \pm 0.041$ | $60.6 \pm 4.5$ |
| Plans + Insights | IPD | $1.379 \pm 0.180$ | $1.169 \pm 0.195$ | $72.3 \pm 3.2$ |
| Reflexion | IPD | $1.403 \pm 0.156$ | $1.207 \pm 0.209$ | $69.5 \pm 2.9$ |
| Social QA | IPD | $1.569 \pm 0.132$ | $1.338 \pm 0.234$ | $68.2 \pm 3.9$ |
| Social LM | IPD | $1.098 \pm 0.089$ | $0.110 \pm 0.037$ | $97.4 \pm 0.8$ |

Table 2: Comparing literal and functional metrics across prompting strategies on Rock, Paper, Scissors (RPS), Iterated Battle of Sexes (IBS), and the Iterated Prisoner's Dilemma (IPD) for Mixtral 8x7B Instruct v1 playing with single action partners.

tasks. We also add *Oracle Prompting* where the actual action the partner will take (and not just a prediction) is directly provided as input (Figure 15). As well as *Oracle + Max Prompting* where maximizing the reward in response to their action is further emphasized. We additionally consider a variant where the interaction history or the payoffs are removed from the prompt. Finally, we added a variant of CoT Prompting where three in-context examples of ideal thought processes are provided (Figure 13) that we call *CoT 3-Shot*.

In Table 3 we provide the regret per step of each prompting strategy across the three games. LLAMA-3 seems to consistently outperform LLAMA-2, but even still Social Prompting does not close the gap with the tabular model. Meanwhile, CoT leads to big improvements sometimes, but is inconsistent, making performance worse for same cases.

> **Gap Remains with Oracle Inputs**
>
> In Table 3 even when the actual actions and payoff structure are given as input, the LLM does not perform effective rational reasoning with this input.

Indeed, it is surprising to see that Oracle consistently performs worse than the tabular RMax model that is learned from scratch without access to the payoff table or knowledge of the other agent's policy. This speaks to a difficulty reasoning over the long contexts of this task. Incentivising maximization in the prompt definitely improves performance, but does not change the overall picture.

For RPS and IBS the interaction history seems vital for per-

formance and the payoff table is not, implying the LLM must strugle to effectively reason about the payoffs. For IPD the combo of both the payoff table and interaction history is destructive such that it does much better with either in isolation. Seeing that the difficulty of reasoning over long contexts is at the core of the issue, we also tried a popular approach called System 2 Attention (S2A) (Weston & Sukhbaatar, 2023) to summarize the payoffs and history before sending it to either QA or CoT Prompting (Figures 16 and 17). S2A adds some value to CoT for IPD, but it is not as good as CoT for RPS and IBS. Still no LLM matches tabular performance.

**Deeper Analysis.** We have conducted a number of experiments that we do not have space to recap in detail within the main text. In Appendix C.3 we validate the generality of our findings regarding functional theory of mind performance across open source models including LLAMA-3 70B Instruct, Mixtral 8x7B Instruct v1, and Mistral Large 2 when paired with more dynamic tit-for-tat style partner policies. Additionally, in Appendix C.4 we look at the role of inductive bias in the action representation on performance of LLAMA-3 70B Instruct and Mistral Large 2 models. We find that more inductive bias is helpful with a small number of interactions, but could lead to poor long-term behavior that does not converge with respect to the partner's policy.

**Reasoning Models.** As our results suggest that achieving functional theory of mind seems related to reasoning over long interaction histories, it is natural to question how the recent emergence of trained chain of thought reasoning models with verifiable rewards such as DeepSeek-R1 (Shao et al.,

| Prompting | RPS | IBS | IPD |
|---|---|---|---|
| Tabular | **0.083** $\pm$ 0.004 | **0.211** $\pm$ 0.012 | **0.086** $\pm$ 0.009 |
| QA | 0.444 $\pm$ 0.107 | 1.391 $\pm$ 0.283 | 0.996 $\pm$ 0.133 |
| CoT | 0.213 $\pm$ 0.015 | 2.475 $\pm$ 0.208 | 0.892 $\pm$ 0.210 |
| CoT + 3-Shot | 0.121 $\pm$ 0.017 | 0.526 $\pm$ 0.081 | 2.773 $\pm$ 0.730 |
| S2A | 0.224 $\pm$ 0.027 | 1.855 $\pm$ 0.314 | 0.808 $\pm$ 0.192 |
| S2A + CoT | 0.234 $\pm$ 0.014 | 2.030 $\pm$ 0.287 | 0.679 $\pm$ 0.137 |
| Social QA | 0.256 $\pm$ 0.017 | 1.613 $\pm$ 0.163 | 0.550 $\pm$ 0.071 |
| Social LM | 0.378 $\pm$ 0.052 | 3.437 $\pm$ 0.154 | 0.803 $\pm$ 0.097 |
| Plans + Insights | 0.173 $\pm$ 0.026 | 2.405 $\pm$ 0.216 | 0.711 $\pm$ 0.114 |
| Reflexion | 0.465 $\pm$ 0.021 | 3.348 $\pm$ 0.105 | 1.076 $\pm$ 0.139 |
| Oracle | 0.238 $\pm$ 0.046 | 1.343 $\pm$ 0.146 | 0.839 $\pm$ 0.056 |
| Oracle + Max | 0.153 $\pm$ 0.014 | 0.785 $\pm$ 0.072 | 0.767 $\pm$ 0.100 |
| Oracle +Max -History | 1.275 $\pm$ 0.152 | 2.060 $\pm$ 0.079 | 0.206 $\pm$ 0.018 |
| Oracle +Max -Payoffs | 0.103 $\pm$ 0.015 | 0.883 $\pm$ 0.097 | 0.241 $\pm$ 0.046 |

Table 3: Ablating long-context reasoning in terms of $\Delta_{\text{Functional}}/T$ for LLAMA-3 70B Instruct across Rock, Paper, Scissors (RPS), Iterated Battle of Sexes (IBS), and the Iterated Prisoner's Dilemma (IPD) when playing with single action partners.

2024) has impacted theory of mind in these models. In Table 4 we compare the performance of the DeepSeek-R1-Distill-Qwen-32B with the Tabular RMax baseline when using the canonical action names for each game. We do indeed see the strongest functional theory of mind performance that we have seen from an LLM thus far when playing with single action partners. It is competitive with or even exceeding the performance of RMax in all games with single action partners. That said, performance is not as consistent with tit-for-tat style dynamic partners. Notably, we also see a bizarre and unique trend in which a rational response to its literal theory of mind predictions is consistently much poorer than the actual behavior generated in the cases when the functional theory of mind performance is impressive.

> **Functional ToM without Literal ToM**
>
> When the trained reasoning model in Table 4 demonstrates strong functional theory of mind, it puzzlingly exceeds its literal theory of mind capabilities.

These results serve to strengthen our case that functional theory of mind and literal theory of mind performance do not directly imply each other. They also provoke interesting questions about the prediction capabilities that may be lost during the training of reasoning models.

## 5. Alternative Views

In this section, we take some time to discuss and refute important alternative views to the position of our paper.

**We should embrace game theory, not theory of mind.** This is perhaps the most prominent alternative view to our paper that was already highlighted in our Rock, Paper, Scissors example. Many papers in the multi-agent RL literature consider finding a game theoretic equilibrium solution as

their ultimate goal (Littman et al., 2001; Wang & Sandholm, 2002; Greenwald et al., 2003; Zinkevich et al., 2005; 2007). In some ways game theoretic solutions are opposite to solutions that are found with theory of mind because they focus on performance against worst case or optimal partners rather than the partners agents they are actually paired with. Likewise, they focus on eliminating the exploitability of solutions by other agents as opposed to maximizing expected return in the context of the actual policies of these agents. Proponents of the game theoretic approach will readily acknowledge that theory of mind is more descriptive of the way that humans tend to act (Colman, 2003; Larson, 2004; Owen, 2013). The argument is rather that game theory is prescriptive of the way that AI agents should ideally act. We believe that there is indeed some validity to this point and acknowledge that in a closed system in which all involved agents are AI, this is the optimal case. However, this does not hold as the community moves towards increasingly real-world use cases in which AI must interact with humans or AI agents that behave sub-optimally. These use cases are the norm for LLMs and are the setting in which researchers are interested in measuring LLM theory of mind capabilities. Here there is a gap between the expected reward achievable with the theory of mind solution that optimizes for the expected case and the game theoretic solution that optimizes for the worst case. Moreover, this gap should only get larger as the agent interacts with more humans. We definitely acknowledge that what society will end up preferring depends on the alignment of incentive structures. Humans will probably prefer theory of mind solutions when their relationship with the agent is mostly collaborative, but will not want to be exploited when their relationship with the agent is mostly adversarial. We strongly support instituting guardrails in these cases and believe that this makes it even more important to measure functional theory of mind performance to understand the extent of exploitation by AI models. We provide further

| Model | Game | Partner | $\Delta_{\text{Functional}}/T$ | $\Delta_{\text{ToM}}/T$ | ToM % |
|---|---|---|---|---|---|
| Tabular | RPS | Single Action | $0.083 \pm 0.004$ | $0.039 \pm 0.006$ | $97.4 \pm 0.4$ |
| DeepSeek-R1 Distilled 32B | RPS | Single Action | $0.074 \pm 0.009$ | $0.544 \pm 0.126$ | $63.7 \pm 8.4$ |
| Tabular | RPS | Tit-For-Tat | $0.224 \pm 0.007$ | $0.105 \pm 0.000$ | $93.0 \pm 0.0$ |
| DeepSeek-R1 Distilled 32B | RPS | Tit-For-Tat | $0.906 \pm 0.041$ | $0.921 \pm 0.029$ | $38.6 \pm 2.0$ |
| Tabular | IBS | Single Action | $0.211 \pm 0.012$ | $0.088 \pm 0.020$ | $98.7 \pm 0.3$ |
| DeepSeek-R1 Distilled 32B | IBS | Single Action | $0.126 \pm 0.045$ | $0.233 \pm 0.078$ | $97.2 \pm 0.8$ |
| Tabular | IBS | Tit-For-Tat | $0.468 \pm 0.031$ | $0.162 \pm 0.005$ | $98.1 \pm 0.1$ |
| DeepSeek-R1 Distilled 32B | IBS | Tit-For-Tat | $0.045 \pm 0.011$ | $0.466 \pm 0.065$ | $94.5 \pm 0.8$ |
| Tabular | IPD | Single Action | $0.086 \pm 0.009$ | $0.071 \pm 0.015$ | $98.6 \pm 0.3$ |
| DeepSeek-R1 Distilled 32B | IPD | Single Action | $0.121 \pm 0.014$ | $0.360 \pm 0.111$ | $83.1 \pm 5.8$ |
| Tabular | IPD | Tit-For-Tat | $0.248 \pm 0.005$ | $0.070 \pm 0.000$ | $98.0 \pm 0.0$ |
| DeepSeek-R1 Distilled 32B | IPD | Tit-For-Tat | $4.789 \pm 0.061$ | $0.945 \pm 0.139$ | $88.9 \pm 1.6$ |

Table 4: Comparing literal and functional metrics on Rock, Paper, Scissors (RPS), Iterated Battle of Sexes (IBS), and the Iterated Prisoner's Dilemma (IPD) for DeepSeek-R1-Distill-Qwen-32B playing with single action partners.

detail about the connection between functional theory of mind and relevant equilibria concepts in Appendix B.

**Literal theory of mind is paramount for insights related applications.** For some applications, we only desire that AI agents provide us with insights, which we will use to make decisions. For these applications, we do not wish to provide the AI with the agency to make these decisions for us. There are indeed very important use cases of this type for which literal theory of mind would be a sufficient capability without functional theory of mind. Generally, these applications are most helpful when the AI has access to more information than the human pertaining to the situation. However, often times the amount of agency we decide to give to an AI depends on its expected reliability in making actual decisions. In this paper we argue that functional theory of mind serves as a necessary test for deciding if AI is ready to receive this agency and that literal theory of mind on its own is not reliable for assessing this performance. It is also important to note that in most applications where we do not provide AI with the agency to act, it would still be useful to humans to have the AI provide an action recommendation.

**The word "broken" is too strong.** It can be argued that because there are many ways to measure literal theory of mind, adding functional theory of mind is a just single additional aspect within a comprehensive theory of mind evaluation. However, we think "broken" is an appropriate term here because functional theory of mind is what we really care about for applications where we give the AI agency to act, and we have found that deploying an LLM in this setting simply based on its literal theory of mind performance can be misleading to a potentially dangerous extent.

**Functional theory of mind is easy with fine-tuning.** RL style fine-tuning would easily solve the functional theory of mind problem displayed in our matrix game based experiments as highlighted with simple tabular models. However, our experiments still demonstrate a case where LLM literal

theory of mind capabilities do not imply functional theory of mind capabilities and thus highlight the importance of directly measuring functional theory of mind. Most prominent LLM based services today also do not actually tune the model weights separately for each user and rely on in-context learning. The likely reason for this is partly about hardware related restrictions and partly about the difficulty of performing reliable continual learning. See Appendix B for a detailed discussion of connections to continual learning.

**The "predict-then-optimize" perspective.** Our paper is also related to the end-to-end predict-and-optimize and two-stage predict-then-optimize literature (Elmachtoub & Grigas, 2022). End-to-end approaches that jointly optimize prediction and decision-making often outperform two-stage approaches where prediction is performed independently from optimization because not all prediction errors are equally consequential for the final decision quality. This perspective nicely highlights how analyzing *literal theory of mind* can lead to misleading conclusions about true *functional theory of mind* performance, even with ideal self-consistent models.

## 6. Conclusion

In this paper, we have argued that existing theory of mind benchmarks are broken because they measure literal theory of mind (Definition 2.1), and not functional theory of mind (Definition 2.2), which is what we typically actually care about in multi-agent social contexts. In Section 4 and Appendix C we provided comprehensive evidence that functional theory of mind performance can be quite poor even when literal theory of mind performance is good and vice versa. Our analysis considers LLMs playing simple matrix games with very simple partners to underscore this point. This position paper serves as a call to action to develop suitably complex LLM theory of mind benchmarks that directly measure functional theory of mind when acting with partners from different personas and social contexts.

## Acknowledgements

MR acknowledges support from the Canada Excellence Research Chairs (CERC) Program. We are grateful for the valuable feedback we received from the ICML 2025 reviewers, which led to substantial improvements to our manuscript. Finally, we would like to thank Tim Klinger and Francesca Rossi for their valuable feedback regarding the use of the term "theory of mind" in other fields that helped shape the way we position our view in Section 1.

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

## A. Literal Theory of Mind and Functional Theory of Mind in Popular Theory of Mind Games

In this section, we provide an explanation of how literal theory of mind and functional theory of mind materialize in popular theory of mind focused games. We refer the reader to the cited papers for full descriptions of how each game is played.

**Codenames (Bills et al., 2025).** In Codenames, literal theory of mind could naturally be measured in terms of guesses by the guesser about the board-card assignments. These assignment can be retrieved from some transformation of the actions that the clue giver would take in different game states following Definition 2.1. Functional theory of mind would be the guesser's regret with respect to the actual team performance on the game following Definition 2.2, which also includes an aspect of reasoning on top of a literal theory of mind model. For example, given the context of the game, guessers may decide to save certain guesses for later to avoid the potential negative consequences of guessing wrong and potentially gifting the other team an opportunity. We argue in this paper, that it is quite possible that a Codenames agent could perform well in terms of literal theory of mind on this kind of theory of mind task while simultaneously struggling to turn that into a rational strategy, displaying poor functional theory of mind as a result.

**Hanabi (Bard et al., 2020).** In Hanabi, literal theory of mind could be measured in terms of each agent's belief about what their own cards are. This belief can be retrieved from some transformation of the actions that other agents took in different game states following Definition 2.1. Functional theory of mind would be the regret with respect to the team score actually achieved at the end of the game. Thus achieving functional theory of mind requires reasoning on top of the theory of mind model regarding how the agent's cards effects the best actions for the agent to take at various points in the game. It is once again quite possible that a Hanabi agent could perform well in terms of literal theory of mind on this kind of theory of mind task while simultaneously struggling to turn that into a rational strategy in response.

**Taboo (Ashktorab et al., 2020).** In Taboo, literal theory of mind could be measured in terms of the clue giver's understanding of the strength of word associations for the particular guesser they are playing with. These associations can be retrieved from some transformation of the actions that the guesser has taken in different previous situations following Definition 2.1 (inside of or outside of the context of the game). Functional theory of mind would be the regret with respect to the team score actually achieved at the end of the game. This again requires additional reasoning on top of the literal theory of mind model, particularly because of the banned list of words that the clue giver cannot say. Thus the clue giver may be inefficient in their search through the space of association strengths even if they have a strong literal theory of mind model.

**Wavelength (Morrison et al., 2025).** In Wavelength, literal theory of mind could be measured similarly to Taboo in terms of understanding the strength of word associations for the particular partner they are playing with. These associations can be retrieved from some transformation of the actions that the guesser has taken in different previous situations following Definition 2.1 (inside of or outside of the context of the game). Functional theory of mind would again be the regret with respect to the team score actually achieved at the end of the game. The additional reasoning required in this case for achieving functional theory of mind is an understanding of how the other agent thinks about the spectrum dial calibration. Thus the agent can still guess wrong regarding the spectrum even if have a perfect understanding of the other agent's associations regarding the clue phrase.

## B. Connections to Equilibria Concepts, Continual Learning, and Potential Applications

**Functional Theory of Mind Objective vs. Equilibria.** The distinction between each agent maximizing their own objective with no control over the optimality of other agents and an equilibria is discussed in detail in (Kim et al., 2022a). In the terminology of that paper, each agent can maximize their search over self-stable periodic distributions of states and rewards, but it only can be considered an equilibria if every agent in the environment simultaneously achieves an optimal policy with respect to its context. Depending on the dynamism of the LLM policies, different classes of equilibria concepts may occur including Nash, Cyclic, Correlated, or Active variants (Kim et al., 2022b). In practice, the functional theory of mind objective with in-context learning matches that of a multi-agent meta-learning policy in order to adapt to an encountered distribution of other agents as in (Kim et al., 2021).

**In-Context Learning vs. Continual Learning.** In a multi-agent environment, the environment is nonstationary from the perspective of each agent if the policies of the other agents it interacts with are changing or learning over time (Littman, 1994). As a result, the setting of rapid adaptation to new agents in our paper can be considered a special case of continual RL (see Proposition 3 of Khetarpal et al. (2020)). Proposition 2 of Khetarpal et al. (2020) established that all continual RL problems can be modeled as partial observable problems with agents conditioned on the full interaction history. So, in principle, in-context learning over sufficiently long context representations with sufficiently expressive neural networks

can be considered a general solution to continual learning problems. However, it remains to be seen if it is possible to see in-context behaviors resembling continual learning strategies like reservoir sampling replay buffers (Riemer et al., 2018a) or scalable memory efficient approximate buffers (Riemer et al., 2017a; 2019; Bashivan et al., 2019). It would be interesting to also consider policy changes from step to step and their degree of correspondence with older models similar to work leveraging knowledge distillation for continual learning (Li & Hoiem, 2017; Riemer et al., 2017b; Kirkpatrick et al., 2017). Intuitively, chain of thought reasoning may be beneficial because of its similarity with continual learning approaches that can select which layers to process at inference time (Rosenbaum et al., 2018; Cases et al., 2019; Chang et al., 2019; Rosenbaum et al., 2019a), see (Rosenbaum et al., 2019b) for a survey of approaches. More generally, this adaptive computation problem can be formalized within the coagent networks framework (Thomas, 2011; Kostas et al., 2020; Zini et al., 2020). That said, the compositional generalization needed for utilizing these models in practice makes achieving real-world success very challenging (Klinger et al., 2020), potentially making LLM style training a more viable strategy for learning this composition. That said, a word of caution comes from recent theoretical insights about the difficulty of efficiently evaluating models with large context lengths (Riemer et al., 2024a) acting in complex environments (Riemer et al., 2022). This could be a leading reason for the poor performance we are currently seeing with LLMs. Ultimately, there are many potentially interesting continual learning settings that should be considered for a comprehensive evaluation (Normandin et al., 2021), which we leave to future work.

**Alternatives to Continual Training.** While our work highlights the limitations of using in-context learning for continual learning with current LLMs, full fine-tuning or continual training can be quite expensive in the context of LLMs – especially when it must be done for each end user. However, more computationally efficient alternatives like parameter efficient tuning (Hu et al., 2021; Dettmers et al., 2024; Thakkar et al., 2024a) or model merging (Ilharco et al., 2022; Yadav et al., 2024; Akiba et al., 2024; Thakkar et al., 2024b) may constitute a more economical middle ground. We leave exploration of the comparative efficacy of these strategies to future work. It is important to note that neural scaling laws only demonstrate generalization improvements with bigger model sizes when there is access to as much data as needed (Kaplan et al., 2020). In fact, capacity limits have been found to enhance generalization for RL (Malloy et al., 2020b; 2023) and multi-agent RL (Malloy et al., 2020a; 2021b;a) in the limited data regime. Smaller models also may be necessary to maintain performance in realtime environments that are sufficiently stochastic (Riemer et al., 2024c;b). As such, it may make more sense to consider smaller LLMs if we plan on doing user specific finetuning or auxiliary objectives we would like to maintain like moral values (Padhi et al., 2024; Dognin et al., 2024).

**Applicability Beyond Multi-agent RL.** In this paper, we focused on multi-agent RL environments because of our use of this formalism in Definitions 2.1 and 2.2. However, the principles discussed in this paper should apply to many practical domains that are not typically modeled using RL such as biomedical applications (Das et al., 2018), making decisions based on conversation topics across the internet (Riemer et al., 2015; Heath et al., 2015; Riemer, 2017; Khabiri et al., 2015), learning what information to teach other models (Omidshafiei et al., 2019), and making decisions based on incoming internet data (Riemer et al., 2016). These problems can generally be considered a special case of the RL formalism (Barto & Dietterich, 2004).

# C. Experiment Details

For our experiments we ran each LLM model using the Watsonx API playing a sequence of 100 step episodes with selected policies for the other agents over the course of 24 hours or a maximum of 100 episodes. Every model was evaluated for at least 30 episodes and confidence intervals are based on the actual sample size considered for each LLM model. The payoff tables used for each game are provided in Tables 5, 6, and 7. We also considered the following representations for each action as indicated in the main text:

- RPS
    - $action_0$: *R*, *J*, *J J J J J J J J J J J J J J J J J J J J*, Rock, Pasta
    - $action_1$: *P*, *F*, *F F F F F F F F F F F F F F F F F F F F*, Paper, Rice
    - $action_2$: *S*, *B*, *B B B B B B B B B B B B B B B B B B B B*, Scissors, Bread

- IBS
    - $action_0$: *J*, *J J J J J J J J J J J J J J J J J J J J*, Fight, Pasta
    - $action_1$: *F*, *F F F F F F F F F F F F F F F F F F F F*, Ballet, Rice

|  |  | Partner | | |
|---|---|---|---|---|
|  |  | $action_0$ | $action_1$ | $action_2$ |
| You | $action_0$ | $(0, 0)$ | $(-1, +1)$ | $(+1, -1)$ |
|  | $action_1$ | $(+1, -1)$ | $(0, 0)$ | $(-1, +1)$ |
|  | $action_2$ | $(-1, +1)$ | $(+1, -1)$ | $(0, 0)$ |

Table 5: Rock, Paper, Scissors (RPS) Payoff Table

|  |  | Partner | |
|---|---|---|---|
|  |  | $action_0$ | $action_1$ |
| You | $action_0$ | $(10, 7)$ | $(0, 0)$ |
|  | $action_1$ | $(0, 0)$ | $(7, 10)$ |

Table 6: Iterated Batte of Sexes (IBS) Payoff Table

- IPD
    - $action_0$: $J$, $JJJJJJJJJJJJJJJJJJJJ$, Cooperate, Pasta
    - $action_1$: $F$, $FFFFFFFFFFFFFFFFFFFF$, Defect, Rice

In Tables 1 and 2, the names $R$, $P$, and $S$ are used for $action_0$, $action_1$, and $action_2$ respectively as we only consider the RPS game. The LLAMA-2 70B Chat model is also quantized for runtime efficiency. In Tables 3 and 10 the names $J$, $F$, and $B$ are used for $action_0$, $action_1$, and $action_2$ respectively to promote consistency with the action spaces also used in IBS and IPD.

## C.1. Example Prompts

In this section, we provide example prompts for each prompting type. In all cases, we tailor our provided example to the Iterated Battle of Sexes (IBS) with actions $J$ and $F$ in which the agent happens to be playing round 5 of 100. There is a slight change in terminology for Rock, Paper, Scissors (RPS) where it is said that the agents "receive a score of +1/0/-1" rather than that they "win 10/8/7/5/0 points". This is just to accommodate for a negative scale of rewards in this game as we did not feel having the reward as always positive would properly reflect the incentive structure of the game to the LLMs.

## C.2. Additional Details on Experiments with Single Action Partners

The best memory size for Reflexion was 1 for RPS, IBS, and IPD in Table 2. In Table 3, the best memory sizes for Reflexion was 1 for RPS and IBS, but it was 3 for IPD.

We provide comprehensive results to augment the analysis of Table 2 in Tables 8 and 9.

## C.3. Additional Experiments with Simple Adaptive Partners

Our main aim is to build towards LLMs that display mutual theory of mind capabilities (Wang & Goel, 2022) in which theory of mind is used to foster coordination behavior between agents. As such, the single action agents may not be realistic as they do not shift their actions in response to the LLM agent's actions. In Table 10 we test the prompting strategies we have considered previously, but now playing with tit for tat style strategies (Axelrod, 1980) made famous for the efficacy in

|  |  | Partner | |
|---|---|---|---|
|  |  | $action_0$ | $action_1$ |
| You | $action_0$ | $(8, 8)$ | $(0, 10)$ |
|  | $action_1$ | $(10, 0)$ | $(5, 5)$ |

Table 7: Iterated Prisoner's Dilemma (IPD) Payoff Table

You are playing a game repeatedly with another player. In this game, you can choose between Option J and Option F. You will play 100 rounds in total with the same player.

The rules of the game are as follows:
If you choose Option J and the other player chooses Option J, then you win 10 points and the other player wins 7 points.
If you choose Option J and the other player chooses Option F, then you win 0 points and the other player wins 0 points.
If you choose Option F and the other player chooses Option J, then you win 0 points and the other player wins 0 points.
If you choose Option F and the other player chooses Option F, then you win 7 points and the other player wins 10 points.

In round 1, you chose Option J and the other player chose Option J. Thus, you won 10 points and the other player won 7 points.
In round 2, you chose Option F and the other player chose Option J. Thus, you won 0 points and the other player won 0 points.
In round 3, you chose Option J and the other player chose Option J. Thus, you won 10 points and the other player won 7 points.
In round 4, you chose Option J and the other player chose Option J. Thus, you won 10 points and the other player won 7 points.
You are currently playing round 5.
Q: Which Option do you choose, Option J or Option F?
A: Option {action}

Figure 1: **LM Prompting Example**. The LLM is given this prompt for each action for the respective game $action_0$, $action_1$, and $action_2$ (for RPS only) and we compare its probabilities computed for the token representation of the actions to determine which action it selects.

You are playing a game repeatedly with another player. In this game, you can choose between Option J and Option F. You will play 100 rounds in total with the same player.

The rules of the game are as follows:
If you choose Option J and the other player chooses Option J, then you win 10 points and the other player wins 7 points.
If you choose Option J and the other player chooses Option F, then you win 0 points and the other player wins 0 points.
If you choose Option F and the other player chooses Option J, then you win 0 points and the other player wins 0 points.
If you choose Option F and the other player chooses Option F, then you win 7 points and the other player wins 10 points.

In round 1, you chose Option J and the other player chose Option J. Thus, you won 10 points and the other player won 7 points.
In round 2, you chose Option F and the other player chose Option J. Thus, you won 0 points and the other player won 0 points.
In round 3, you chose Option J and the other player chose Option J. Thus, you won 10 points and the other player won 7 points.
In round 4, you chose Option J and the other player chose Option J. Thus, you won 10 points and the other player won 7 points.
In round 5, you chose Option J and the other player chose Option {action}

Figure 2: **LM Literal Theory of Mind Prompting - Agent First**. The LLM is given this prompt for each action that its partner will take for the respective game $action_0$, $action_1$, and $action_2$ (for RPS only) and we compare its probabilities computed for the token representation of the actions to determine which action prediction it selects.

You are playing a game repeatedly with another player. In this game, you can choose between Option J and Option F. You will play 100 rounds in total with the same player.

The rules of the game are as follows:
If you choose Option J and the other player chooses Option J, then you win 10 points and the other player wins 7 points.
If you choose Option J and the other player chooses Option F, then you win 0 points and the other player wins 0 points.
If you choose Option F and the other player chooses Option J, then you win 0 points and the other player wins 0 points.
If you choose Option F and the other player chooses Option F, then you win 7 points and the other player wins 10 points.

In round 1, the other player chose Option J and you chose Option J. Thus, you won 10 points and the other player won 7 points.
In round 2, the other player chose Option J and you chose Option F. Thus, you won 0 points and the other player won 0 points.
In round 3, the other player chose Option J and you chose Option J. Thus, you won 10 points and the other player won 7 points.
In round 4, the other player chose Option J and you chose Option J. Thus, you won 10 points and the other player won 7 points.
In round 5, the other player chose Option {action}

Figure 3: **LM Literal Theory of Mind Prompting - Partner First**. The LLM is given this prompt for each action that its partner will take for the respective game $action_0$, $action_1$, and $action_2$ (for RPS only) and we compare its probabilities computed for the token representation of the actions to determine which action prediction it selects.

You are playing a game repeatedly with another player. In this game, you can choose between Option J and Option F. You will play 100 rounds in total with the same player.

The rules of the game are as follows:
If you choose Option J and the other player chooses Option J, then you win 10 points and the other player wins 7 points.
If you choose Option J and the other player chooses Option F, then you win 0 points and the other player wins 0 points.
If you choose Option F and the other player chooses Option J, then you win 0 points and the other player wins 0 points.
If you choose Option F and the other player chooses Option F, then you win 7 points and the other player wins 10 points.

In round 1, you chose Option J and the other player chose Option J. Thus, you won 10 points and the other player won 7 points.
In round 2, you chose Option F and the other player chose Option J. Thus, you won 0 points and the other player won 0 points.
In round 3, you chose Option J and the other player chose Option J. Thus, you won 10 points and the other player won 7 points.
In round 4, you chose Option J and the other player chose Option J. Thus, you won 10 points and the other player won 7 points.
You are currently playing round 5.
Which Option do you choose, Option J or Option F?
Your answer MUST be formatted like:
Option: <J or F>

Figure 4: **QA Prompting**. The LLM is given this prompt and generates an answer with stochastic decoding. If the provided response matches the template and includes an action from the action list, the response is accepted. If not, we query the LLM for more stochastic responses until the criteria is satisfied. Failures were relatively uncommon.

You are playing a game repeatedly with another player. In this game, you can choose between Option J and Option F. You will play 100 rounds in total with the same player.

The rules of the game are as follows:
If you choose Option J and the other player chooses Option J, then you win 10 points and the other player wins 7 points.
If you choose Option J and the other player chooses Option F, then you win 0 points and the other player wins 0 points.
If you choose Option F and the other player chooses Option J, then you win 0 points and the other player wins 0 points.
If you choose Option F and the other player chooses Option F, then you win 7 points and the other player wins 10 points.

In round 1, you chose Option J and the other player chose Option J. Thus, you won 10 points and the other player won 7 points.
In round 2, you chose Option F and the other player chose Option J. Thus, you won 0 points and the other player won 0 points.
In round 3, you chose Option J and the other player chose Option J. Thus, you won 10 points and the other player won 7 points.
In round 4, you chose Option J and the other player chose Option J. Thus, you won 10 points and the other player won 7 points.
In round 5, you chose Option J.
Which Option did you think the other player chose?
Your answer MUST be formatted like:
Option: <J or F>

Figure 5: **QA Literal Theory of Mind Prompting**. The LLM is given this prompt and generates an answer with stochastic decoding. If the provided response matches the template and includes an action from the action list, the response is accepted. If not, we query the LLM for more stochastic responses until the criteria is satisfied. Failures were relatively uncommon.

You are playing a game repeatedly with another player. In this game, you can choose between Option J and Option F. You will play 100 rounds in total with the same player.

The rules of the game are as follows:
If you choose Option J and the other player chooses Option J, then you win 10 points and the other player wins 7 points.
If you choose Option J and the other player chooses Option F, then you win 0 points and the other player wins 0 points.
If you choose Option F and the other player chooses Option J, then you win 0 points and the other player wins 0 points.
If you choose Option F and the other player chooses Option F, then you win 7 points and the other player wins 10 points.

In round 1, you chose Option J and the other player chose Option J. Thus, you won 10 points and the other player won 7 points.
In round 2, you chose Option F and the other player chose Option J. Thus, you won 0 points and the other player won 0 points.
In round 3, you chose Option J and the other player chose Option J. Thus, you won 10 points and the other player won 7 points.
In round 4, you chose Option J and the other player chose Option J. Thus, you won 10 points and the other player won 7 points.
You are currently playing round 5.
Which Option do you choose, Option J or Option F?
Choose the best action by thinking step by step. Your answer MUST be formatted as:
Thoughts: <paragraph explaining your reasoning>
Option: <J or F>

Figure 6: **CoT Prompting**. The LLM is given this prompt and generates an answer with stochastic decoding. If the provided response matches the template and includes an action from the action list, the response is accepted. If not, we query the LLM for more stochastic responses until the criteria is satisfied. Failures were relatively uncommon.

You are playing a game repeatedly with another player. In this game, you can choose between Option J and Option F. You will play 100 rounds in total with the same player.

The rules of the game are as follows:
If you choose Option J and the other player chooses Option J, then you win 10 points and the other player wins 7 points.
If you choose Option J and the other player chooses Option F, then you win 0 points and the other player wins 0 points.
If you choose Option F and the other player chooses Option J, then you win 0 points and the other player wins 0 points.
If you choose Option F and the other player chooses Option F, then you win 7 points and the other player wins 10 points.

In round 1, you chose Option J and the other player chose Option J. Thus, you won 10 points and the other player won 7 points.
In round 2, you chose Option F and the other player chose Option J. Thus, you won 0 points and the other player won 0 points.
In round 3, you chose Option J and the other player chose Option J. Thus, you won 10 points and the other player won 7 points.
In round 4, you chose Option J and the other player chose Option J. Thus, you won 10 points and the other player won 7 points.
In round 5, you chose Option J.
Which Option do you think the other player chose? Choose the best action by thinking step by step. Your answer MUST be formatted as:
Thoughts: <paragraph explaining your reasoning>
Option: <J or F>

Figure 7: **CoT Literal Theory of Mind Prompting**. The LLM is given this prompt and generates an answer with stochastic decoding. If the provided response matches the template and includes an action from the action list, the response is accepted. If not, we query the LLM for more stochastic responses until the criteria is satisfied. Failures were relatively uncommon.

You are playing a game repeatedly with another player. In this game, you can choose between Option J and Option F. You will play 100 rounds in total with the same player.

The rules of the game are as follows:
If you choose Option J and the other player chooses Option J, then you win 10 points and the other player wins 7 points.
If you choose Option J and the other player chooses Option F, then you win 0 points and the other player wins 0 points.
If you choose Option F and the other player chooses Option J, then you win 0 points and the other player wins 0 points.
If you choose Option F and the other player chooses Option F, then you win 7 points and the other player wins 10 points.

In round 1, you chose Option J and the other player chose Option J. Thus, you won 10 points and the other player won 7 points.
In round 2, you chose Option F and the other player chose Option J. Thus, you won 0 points and the other player won 0 points.
In round 3, you chose Option J and the other player chose Option J. Thus, you won 10 points and the other player won 7 points.
In round 4, you chose Option J and the other player chose Option J. Thus, you won 10 points and the other player won 7 points.
Your plans from the last round were: {old_plans}
Your insights from the last round were: {old_insights}
You are currently playing round 5.
Which Option do you choose, Option J or Option F?
Choose the best action by thinking step by step. Your answer MUST be formatted as:
Plans: <paragraph explaining your plan moving forward>
Insights: <paragraph detailing any useful insights you can conclude>
Option: <J or F>

Figure 8: **Plans+Insights Prompting**. The LLM is given this prompt and generates an answer with stochastic decoding. If the provided response matches the template and includes an action from the action list, the response is accepted. If not, we query the LLM for more stochastic responses until the criteria is satisfied. Failures were relatively uncommon.

You are playing a game repeatedly with another player. In this game, you can choose between Option J and Option F. You will play 100 rounds in total with the same player.

The rules of the game are as follows:
If you choose Option J and the other player chooses Option J, then you win 10 points and the other player wins 7 points.
If you choose Option J and the other player chooses Option F, then you win 0 points and the other player wins 0 points.
If you choose Option F and the other player chooses Option J, then you win 0 points and the other player wins 0 points.
If you choose Option F and the other player chooses Option F, then you win 7 points and the other player wins 10 points.

In round 1, you chose Option J and the other player chose Option J. Thus, you won 10 points and the other player won 7 points.
In round 2, you chose Option F and the other player chose Option J. Thus, you won 0 points and the other player won 0 points.
In round 3, you chose Option J and the other player chose Option J. Thus, you won 10 points and the other player won 7 points.
In round 4, you chose Option J and the other player chose Option J. Thus, you won 10 points and the other player won 7 points.
Your plans from this round were: {current_plans}
Your insights from this round were: {current_insights}
In round 5, you chose Option J.
Which Option did you think the other player chose?
Your answer MUST be formatted like:
Option: <J or F>

Figure 9: **Plans+Insights Literal Theory of Mind Prompting**. The LLM is given this prompt and generates an answer with stochastic decoding. If the provided response matches the template and includes an action from the action list, the response is accepted. If not, we query the LLM for more stochastic responses until the criteria is satisfied. Failures were relatively uncommon.

You are playing a game repeatedly with another player. In this game, you can choose between Option J and Option F. You will play 100 rounds in total with the same player.

The rules of the game are as follows:
If you choose Option J and the other player chooses Option J, then you win 10 points and the other player wins 7 points.
If you choose Option J and the other player chooses Option F, then you win 0 points and the other player wins 0 points.
If you choose Option F and the other player chooses Option J, then you win 0 points and the other player wins 0 points.
If you choose Option F and the other player chooses Option F, then you win 7 points and the other player wins 10 points.

In round 1, you chose Option J and the other player chose Option J. Thus, you won 10 points and the other player won 7 points.
In round 2, you chose Option F and the other player chose Option J. Thus, you won 0 points and the other player won 0 points.
In round 3, you chose Option J and the other player chose Option J. Thus, you won 10 points and the other player won 7 points.
In round 4, you chose Option J and the other player chose Option J. Thus, you won 10 points and the other player won 7 points.

Given the history of past experiences above, think about the strategy you implemented and a devise a concise new plan of action that accounts for any mistakes with reference to specific actions that you should have taken. For example, if you tried X and Y but forgot Z, then devise a plan to achieve Z with environment-specific actions. You will need this later when you are solving the same task. Your answer MUST be formatted as:
Plan: <concise explanation of your plan moving forward>"

Figure 10: **Reflection Module Prompting for Reflexion Prompting**. The plan output from this prompt fills in the most recent memory for the Reflexion prompts in Figures 11 and 12.

You are playing a game repeatedly with another player. In this game, you can choose between Option J and Option F. You will play 100 rounds in total with the same player.

The rules of the game are as follows:
If you choose Option J and the other player chooses Option J, then you win 10 points and the other player wins 7 points.
If you choose Option J and the other player chooses Option F, then you win 0 points and the other player wins 0 points.
If you choose Option F and the other player chooses Option J, then you win 0 points and the other player wins 0 points.
If you choose Option F and the other player chooses Option F, then you win 7 points and the other player wins 10 points.

In round 1, you chose Option J and the other player chose Option J. Thus, you won 10 points and the other player won 7 points.
In round 2, you chose Option F and the other player chose Option J. Thus, you won 0 points and the other player won 0 points.
In round 3, you chose Option J and the other player chose Option J. Thus, you won 10 points and the other player won 7 points.
In round 4, you chose Option J and the other player chose Option J. Thus, you won 10 points and the other player won 7 points.
Your plan from three rounds ago was: {old_plan_t-3}
Your plan from two rounds ago was: {old_plan_t-2}
Your plan from the previous round was: {old_plan_t-1}
You are currently playing round 5.
Which Option do you choose, Option J or Option F?
Choose the best action by thinking step by step. Your answer MUST be formatted as:
Thoughts: <paragraph explaining your reasoning>
Option: <J or F>

Figure 11: **Reflexion Prompting (3 Memories)**. The LLM is given this prompt and generates an answer with stochastic decoding. If the provided response matches the template and includes an action from the action list, the response is accepted. If not, we query the LLM for more stochastic responses until the criteria is satisfied. Failures were relatively uncommon.

You are playing a game repeatedly with another player. In this game, you can choose between Option J and Option F. You will play 100 rounds in total with the same player.

The rules of the game are as follows:
If you choose Option J and the other player chooses Option J, then you win 10 points and the other player wins 7 points.
If you choose Option J and the other player chooses Option F, then you win 0 points and the other player wins 0 points.
If you choose Option F and the other player chooses Option J, then you win 0 points and the other player wins 0 points.
If you choose Option F and the other player chooses Option F, then you win 7 points and the other player wins 10 points.

In round 1, you chose Option J and the other player chose Option J. Thus, you won 10 points and the other player won 7 points.
In round 2, you chose Option F and the other player chose Option J. Thus, you won 0 points and the other player won 0 points.
In round 3, you chose Option J and the other player chose Option J. Thus, you won 10 points and the other player won 7 points.
In round 4, you chose Option J and the other player chose Option J. Thus, you won 10 points and the other player won 7 points.
Your plan from three rounds ago was: {old_plan_t-3}
Your plan from two rounds ago was: {old_plan_t-2}
Your plan from the previous round was: {old_plan_t-1}
In round 5, you chose Option J.
Which Option did you think the other player chose?
Your answer MUST be formatted like:
Option: <J or F>

Figure 12: **Reflexion Literal Theory of Mind Prompting (3 Memories)**. The LLM is given this prompt and generates an answer with stochastic decoding. If the provided response matches the template and includes an action from the action list, the response is accepted. If not, we query the LLM for more stochastic responses until the criteria is satisfied. Failures were relatively uncommon.

You are playing a game repeatedly with another player. In this game, you can choose between Option J and Option F. You will play 100 rounds in total with the same player.

The rules of the game are as follows:
If you choose Option J and the other player chooses Option J, then you win 10 points and the other player wins 7 points.
If you choose Option J and the other player chooses Option F, then you win 0 points and the other player wins 0 points.
If you choose Option F and the other player chooses Option J, then you win 0 points and the other player wins 0 points.
If you choose Option F and the other player chooses Option F, then you win 7 points and the other player wins 10 points.

Some example thought processes for superior play in this game include:
Thoughts: The other player has picked Option J is every round so far, so it is logical to assume that they will do so again. If they choose Option J, my best response is Option J. In that case, I will win 10 points.
Thoughts: I chose Option F in the last round, so the other player is likely to choose Option F in response in order to maximize their reward. If they choose Option F, my best response is Option F. In that case, I will win 7 points.
Thoughts: The other player has picked Option J four times in a row, so they are most likely to choose Option J again. If they choose Option J, my best response is Option J. In that case, I will win 10 points.

In round 1, you chose Option J and the other player chose Option J. Thus, you won 10 points and the other player won 7 points.
In round 2, you chose Option F and the other player chose Option J. Thus, you won 0 points and the other player won 0 points.
In round 3, you chose Option J and the other player chose Option J. Thus, you won 10 points and the other player won 7 points.
In round 4, you chose Option J and the other player chose Option J. Thus, you won 10 points and the other player won 7 points.
In round 5, you chose Option J.
Which Option do you think the other player chose? Choose the best action by thinking step by step. Your answer MUST be formatted as:
Thoughts: <paragraph explaining your reasoning>
Option: <J or F>

Figure 13: **CoT 3-Shot Prompting**. The LLM is given this prompt and generates an answer with stochastic decoding. If the provided response matches the template and includes an action from the action list, the response is accepted. If not, we query the LLM for more stochastic responses until the criteria is satisfied. Failures were relatively uncommon. The provided examples were customized to the actions space and payoff structure of each of the three games.

You are playing a game repeatedly with another player. In this game, you can choose between Option J and Option F. You will play 100 rounds in total with the same player.

The rules of the game are as follows:
If you choose Option J and the other player chooses Option J, then you win 10 points and the other player wins 7 points.
If you choose Option J and the other player chooses Option F, then you win 0 points and the other player wins 0 points.
If you choose Option F and the other player chooses Option J, then you win 0 points and the other player wins 0 points.
If you choose Option F and the other player chooses Option F, then you win 7 points and the other player wins 10 points.

In round 1, you chose Option J and the other player chose Option J. Thus, you won 10 points and the other player won 7 points.
In round 2, you chose Option F and the other player chose Option J. Thus, you won 0 points and the other player won 0 points.
In round 3, you chose Option J and the other player chose Option J. Thus, you won 10 points and the other player won 7 points.
In round 4, you chose Option J and the other player chose Option J. Thus, you won 10 points and the other player won 7 points.
You are currently playing round 5.
The other player will choose Option J. Which Option do you choose, Option J or Option F?
Your answer MUST be formatted like:
Option: <J or F>

Figure 14: **Oracle Prompting**. The provided action that the other agent will choose is true 'oracle' knowledge in all cases. The LLM is given this prompt and generates an answer with stochastic decoding. If the provided response matches the template and includes an action from the action list, the response is accepted. If not, we query the LLM for more stochastic responses until the criteria is satisfied. Failures were relatively uncommon.

You are playing a game repeatedly with another player. In this game, you can choose between Option J and Option F. You will play 100 rounds in total with the same player.

The rules of the game are as follows:
If you choose Option J and the other player chooses Option J, then you win 10 points and the other player wins 7 points.
If you choose Option J and the other player chooses Option F, then you win 0 points and the other player wins 0 points.
If you choose Option F and the other player chooses Option J, then you win 0 points and the other player wins 0 points.
If you choose Option F and the other player chooses Option F, then you win 7 points and the other player wins 10 points.

In round 1, you chose Option J and the other player chose Option J. Thus, you won 10 points and the other player won 7 points.
In round 2, you chose Option F and the other player chose Option J. Thus, you won 0 points and the other player won 0 points.
In round 3, you chose Option J and the other player chose Option J. Thus, you won 10 points and the other player won 7 points.
In round 4, you chose Option J and the other player chose Option J. Thus, you won 10 points and the other player won 7 points.
You are currently playing round 5.
The other player will choose Option J. Which Option do you choose in response to maximize your points received, Option J or Option F?
Your answer MUST be formatted like:
Option: <J or F>

Figure 15: **Oracle Prompting Emphasizing Maximization**. The provided action that the other agent will choose is true 'oracle' knowledge in all cases. The LLM is given this prompt and generates an answer with stochastic decoding. If the provided response matches the template and includes an action from the action list, the response is accepted. If not, we query the LLM for more stochastic responses until the criteria is satisfied. Failures were relatively uncommon. The only difference with Figure 15 in the emphasis on maximizing reward in response to the other agent's action.

You are playing a game repeatedly with another player. In this game, you can choose between Option J and Option F. You will play 100 rounds in total with the same player.

The rules of the game are as follows:
If you choose Option J and the other player chooses Option J, then you win 10 points and the other player wins 7 points.
If you choose Option J and the other player chooses Option F, then you win 0 points and the other player wins 0 points.
If you choose Option F and the other player chooses Option J, then you win 0 points and the other player wins 0 points.
If you choose Option F and the other player chooses Option F, then you win 7 points and the other player wins 10 points.

In round 1, you chose Option J and the other player chose Option J. Thus, you won 10 points and the other player won 7 points.
In round 2, you chose Option F and the other player chose Option J. Thus, you won 0 points and the other player won 0 points.
In round 3, you chose Option J and the other player chose Option J. Thus, you won 10 points and the other player won 7 points.
In round 4, you chose Option J and the other player chose Option J. Thus, you won 10 points and the other player won 7 points.
Given the game rules and limited interaction history while playing with the other player described above, extract a concise, unbiased, and self-contained summary that would be useful for deciding which Option you should choose in the next round.

Figure 16: **S2A Prompting**. We accept the LLM's output for this prompt in all cases. This output replaces the rules and history in the subsequent QA or CoT prompt.

You are playing a game repeatedly with another player. In this game, you can choose between Option J and Option F. You will play 100 rounds in total with the same player.

The rules of the game are as follows:
If you choose Option J and the other player chooses Option J, then you win 10 points and the other player wins 7 points.
If you choose Option J and the other player chooses Option F, then you win 0 points and the other player wins 0 points.
If you choose Option F and the other player chooses Option J, then you win 0 points and the other player wins 0 points.
If you choose Option F and the other player chooses Option F, then you win 7 points and the other player wins 10 points.

In round 1, you chose Option J and the other player chose Option J. Thus, you won 10 points and the other player won 7 points.
In round 2, you chose Option F and the other player chose Option J. Thus, you won 0 points and the other player won 0 points.
In round 3, you chose Option J and the other player chose Option J. Thus, you won 10 points and the other player won 7 points.
In round 4, you chose Option J and the other player chose Option J. Thus, you won 10 points and the other player won 7 points.
Given the game rules and limited interaction history while playing with the other player described above, extract a concise, unbiased, and self-contained summary that would be useful for guessing which Option the other player will choose in the next round.

Figure 17: **S2A Literal Theory of Mind Prompting**. We accept the LLM's output for this prompt in all cases. This output replaces the rules and history in the subsequent QA or CoT prompt.

| Model | Prompting | Game | $\Delta_{\text{Functional}}/T$ | $\Delta_{\text{ToM}}/T$ | ToM % |
|---|---|---|---|---|---|
| LLAMA-2 70B Chat | LM | RPS | $0.857 \pm 0.142$ | $0.119 \pm 0.006$ | $92.1 \pm 0.4$ |
| LLAMA-2 70B Chat | QA | RPS | $1.123 \pm 0.236$ | $0.743 \pm 0.106$ | $50.5 \pm 7.1$ |
| LLAMA-2 70B Chat | CoT | RPS | $0.928 \pm 0.133$ | $0.776 \pm 0.081$ | $48.3 \pm 5.4$ |
| LLAMA-3 70B Instruct | QA | RPS | $0.444 \pm 0.107$ | $0.414 \pm 0.027$ | $72.4 \pm 1.8$ |
| LLAMA-3 70B Instruct | CoT | RPS | $0.213 \pm 0.015$ | $0.911 \pm 0.031$ | $39.3 \pm 2.1$ |
| LLAMA-3 70B Instruct | CoT + 3-Shot | RPS | $0.121 \pm 0.017$ | $0.747 \pm 0.042$ | $50.2 \pm 2.8$ |
| LLAMA-3 70B Instruct | S2A | RPS | $0.224 \pm 0.027$ | $0.209 \pm 0.031$ | $86.1 \pm 2.0$ |
| LLAMA-3 70B Instruct | S2A + CoT | RPS | $0.234 \pm 0.014$ | $0.278 \pm 0.025$ | $81.4 \pm 1.6$ |
| LLAMA-3 70B Instruct | Social QA | RPS | $0.256 \pm 0.017$ | $0.548 \pm 0.032$ | $63.5 \pm 2.1$ |
| LLAMA-3 70B Instruct | Social LM | RPS | $0.378 \pm 0.052$ | $0.109 \pm 0.023$ | $93.1 \pm 1.5$ |
| LLAMA-3 70B Instruct | Plans + Insights | RPS | $0.173 \pm 0.026$ | $0.178 \pm 0.015$ | $88.2 \pm 1.0$ |
| LLAMA-3 70B Instruct | Reflexion | RPS | $0.465 \pm 0.021$ | $0.273 \pm 0.019$ | $81.8 \pm 1.3$ |
| LLAMA-2 70B Chat | LM | IBS | $2.316 \pm 0.817$ | $0.496 \pm 0.038$ | $94.4 \pm 0.4$ |
| LLAMA-2 70B Chat | QA | IBS | $2.570 \pm 0.833$ | $2.401 \pm 0.480$ | $68.8 \pm 7.6$ |
| LLAMA-2 70B Chat | CoT | IBS | $3.367 \pm 0.149$ | $2.301 \pm 0.120$ | $71.9 \pm 2.3$ |
| LLAMA-3 70B Instruct | QA | IBS | $1.391 \pm 0.283$ | $1.247 \pm 0.141$ | $84.6 \pm 2.3$ |
| LLAMA-3 70B Instruct | CoT | IBS | $2.475 \pm 0.208$ | $1.835 \pm 0.128$ | $77.7 \pm 2.4$ |
| LLAMA-3 70B Instruct | CoT + 3-Shot | IBS | $0.526 \pm 0.081$ | $0.997 \pm 0.106$ | $88.3 \pm 1.4$ |
| LLAMA-3 70B Instruct | S2A | IBS | $1.855 \pm 0.314$ | $1.156 \pm 0.175$ | $85.3 \pm 2.9$ |
| LLAMA-3 70B Instruct | S2A + CoT | IBS | $2.030 \pm 0.287$ | $1.300 \pm 0.104$ | $83.9 \pm 1.9$ |
| LLAMA-3 70B Instruct | Social QA | IBS | $1.613 \pm 0.163$ | $1.138 \pm 0.104$ | $85.1 \pm 1.7$ |
| LLAMA-3 70B Instruct | Social LM | IBS | $3.437 \pm 0.154$ | $0.795 \pm 0.043$ | $88.9 \pm 2.4$ |
| LLAMA-3 70B Instruct | Plans + Insights | IBS | $2.405 \pm 0.216$ | $0.756 \pm 0.097$ | $90.6 \pm 1.5$ |
| LLAMA-3 70B Instruct | Reflexion | IBS | $3.348 \pm 0.105$ | $1.117 \pm 0.127$ | $85.5 \pm 2.1$ |
| LLAMA-2 70B Chat | LM | IPD | $2.889 \pm 0.422$ | $0.220 \pm 0.034$ | $93.5 \pm 0.3$ |
| LLAMA-2 70B Chat | QA | IPD | $3.025 \pm 0.383$ | $1.665 \pm 0.430$ | $61.5 \pm 7.1$ |
| LLAMA-2 70B Chat | CoT | IPD | $1.839 \pm 0.289$ | $1.393 \pm 0.166$ | $63.8 \pm 1.6$ |
| LLAMA-3 70B Instruct | QA | IPD | $0.996 \pm 0.133$ | $0.690 \pm 0.053$ | $82.3 \pm 3.0$ |
| LLAMA-3 70B Instruct | CoT | IPD | $0.892 \pm 0.210$ | $0.834 \pm 0.056$ | $72.7 \pm 3.2$ |
| LLAMA-3 70B Instruct | CoT + 3-Shot | IPD | $2.773 \pm 0.730$ | $0.524 \pm 0.061$ | $83.8 \pm 2.3$ |
| LLAMA-3 70B Instruct | S2A | IPD | $0.808 \pm 0.192$ | $0.662 \pm 0.200$ | $84.6 \pm 3.3$ |
| LLAMA-3 70B Instruct | S2A + CoT | IPD | $0.679 \pm 0.137$ | $0.639 \pm 0.106$ | $83.1 \pm 1.4$ |
| LLAMA-3 70B Instruct | Social QA | IPD | $0.550 \pm 0.071$ | $0.638 \pm 0.036$ | $78.0 \pm 2.0$ |
| LLAMA-3 70B Instruct | Social LM | IPD | $0.803 \pm 0.097$ | $0.394 \pm 0.065$ | $87.8 \pm 2.2$ |
| LLAMA-3 70B Instruct | Plans + Insights | IPD | $0.711 \pm 0.114$ | $0.499 \pm 0.070$ | $86.7 \pm 1.0$ |
| LLAMA-3 70B Instruct | Reflexion | IPD | $1.076 \pm 0.139$ | $0.428 \pm 0.055$ | $86.8 \pm 0.9$ |

Table 8: Comparing literal and functional metrics across prompting strategies on Rock, Paper, Scissors (RPS), Iterated Battle of Sexes (IBS), and the Iterated Prisoner's Dilemma (IPD) for LLAMA-2 70B Chat and LLAMA-3 70B Instruct playing with single action partners.

You are playing a game repeatedly with another player. In this game, you can choose between Option J and Option F. You will play 100 rounds in total with the same player.

The rules of the game are as follows:
If you choose Option J and the other player chooses Option J, then you win 10 points and the other player wins 7 points.
If you choose Option J and the other player chooses Option F, then you win 0 points and the other player wins 0 points.
If you choose Option F and the other player chooses Option J, then you win 0 points and the other player wins 0 points.
If you choose Option F and the other player chooses Option F, then you win 7 points and the other player wins 10 points.

In round 1, you chose Option J and the other player chose Option J. Thus, you won 10 points and the other player won 7 points.
In round 2, you chose Option F and the other player chose Option J. Thus, you won 0 points and the other player won 0 points.
In round 3, you chose Option J and the other player chose Option J. Thus, you won 10 points and the other player won 7 points.
In round 4, you chose Option J and the other player chose Option J. Thus, you won 10 points and the other player won 7 points.
Given that you predict the other player will choose Option J in round 5, which Option do you think is best to choose for you in this round, Option J or Option F?
Your answer MUST be formatted like:
Option: <J or F>

Figure 18: **Social Prompting**. The actions that it is predicted that the other player will choose is the actual output of a prior application of literal theory of mind oriented prompt to the same LLM model. If the provided response matches the template and includes an action from the action list, the response is accepted. If not, we query the LLM for more stochastic responses until the criteria is satisfied. Failures were relatively uncommon.

| Model | Prompting | Game | $\Delta_{\text{Functional}}/T$ | $\Delta_{\text{ToM}}/T$ | ToM % |
|---|---|---|---|---|---|
| Mistral Large 2 | QA | RPS | $0.181 \pm 0.013$ | $0.133 \pm 0.01$ | $91.1 \pm 0.7$ |
| Mistral Large 2 | S2A | RPS | $0.177 \pm 0.015$ | $0.134 \pm 0.015$ | $91.1 \pm 1.0$ |
| Mistral Large 2 | S2A + CoT | RPS | $0.127 \pm 0.012$ | $0.259 \pm 0.017$ | $82.8 \pm 1.1$ |
| Mistral Large 2 | Social QA | RPS | $0.126 \pm 0.008$ | $0.081 \pm 0.006$ | $94.6 \pm 0.4$ |
| Mistral Large 2 | Reflexion | RPS | $0.156 \pm 0.009$ | $0.050 \pm 0.005$ | $96.7 \pm 0.3$ |
| Mistral Large 2 | QA | IBS | $1.955 \pm 0.103$ | $0.575 \pm 0.055$ | $93.5 \pm 0.5$ |
| Mistral Large 2 | S2A | IBS | $0.923 \pm 0.094$ | $0.497 \pm 0.051$ | $94.2 \pm 0.6$ |
| Mistral Large 2 | S2A + CoT | IBS | $0.498 \pm 0.091$ | $0.300 \pm 0.049$ | $96.3 \pm 0.7$ |
| Mistral Large 2 | Social QA | IBS | $1.724 \pm 0.132$ | $0.42 \pm 0.036$ | $95.0 \pm 0.4$ |
| Mistral Large 2 | Reflexion | IBS | $1.076 \pm 0.058$ | $0.342 \pm 0.031$ | $96.0 \pm 0.4$ |
| Mistral Large 2 | QA | IPD | $0.688 \pm 0.045$ | $0.27 \pm 0.019$ | $90.4 \pm 1.0$ |
| Mistral Large 2 | S2A | IPD | $0.631 \pm 0.052$ | $0.275 \pm 0.042$ | $92.3 \pm 0.7$ |
| Mistral Large 2 | S2A + CoT | IPD | $0.385 \pm 0.037$ | $0.255 \pm 0.027$ | $91.1 \pm 1.1$ |
| Mistral Large 2 | Social QA | IPD | $0.585 \pm 0.062$ | $0.243 \pm 0.018$ | $91.2 \pm 1.1$ |
| Mistral Large 2 | Reflexion | IPD | $0.619 \pm 0.033$ | $0.171 \pm 0.016$ | $94.5 \pm 0.5$ |

Table 9: Comparing literal and functional metrics across prompting strategies on Rock, Paper, Scissors (RPS), Iterated Battle of Sexes (IBS), and the Iterated Prisoner's Dilemma (IPD) for Mistral Large 2 playing with single action partners.

| Model | Prompting | RPS | IBS | IPD |
|---|---|---|---|---|
| Tabular | N/A | **0.211** $\pm$ 0.007 | 0.468 $\pm$ 0.031 | **0.248** $\pm$ 0.005 |
| Mixtral 8x7b Instruct v1 | QA | 1.224 $\pm$ 0.025 | 1.484 $\pm$ 0.109 | 1.074 $\pm$ 0.086 |
| Mixtral 8x7b Instruct v1 | CoT | 1.052 $\pm$ 0.026 | 4.654 $\pm$ 0.222 | 2.604 $\pm$ 0.038 |
| Mixtral 8x7b Instruct v1 | CoT + 3-Shot | 1.067 $\pm$ 0.016 | 2.590 $\pm$ 0.190 | 1.796 $\pm$ 0.113 |
| Mixtral 8x7b Instruct v1 | S2A | 1.065 $\pm$ 0.021 | 1.627 $\pm$ 0.167 | 2.002 $\pm$ 0.099 |
| Mixtral 8x7b Instruct v1 | S2A + CoT | 0.994 $\pm$ 0.026 | 3.640 $\pm$ 0.418 | 2.675 $\pm$ 0.051 |
| Mixtral 8x7b Instruct v1 | Social LM | 1.304 $\pm$ 0.020 | 1.257 $\pm$ 0.100 | 1.674 $\pm$ 0.135 |
| Mixtral 8x7b Instruct v1 | Plans/Insights* | 1.538 $\pm$ 0.023 | 2.313 $\pm$ 0.195 | 1.791 $\pm$ 0.112 |
| Mixtral 8x7b Instruct v1 | Reflexion 1 | 1.03 $\pm$ 0.023 | 4.169 $\pm$ 0.255 | 2.383 $\pm$ 0.059 |
| Mixtral 8x7b Instruct v1 | Reflexion 3 | 1.076 $\pm$ 0.015 | 3.951 $\pm$ 0.235 | 2.213 $\pm$ 0.064 |
| Mistral Large 2 | QA | 1.064 $\pm$ 0.018 | 2.167 $\pm$ 0.131 | 2.581 $\pm$ 0.048 |
| Mistral Large 2 | CoT | 1.007 $\pm$ 0.014 | 0.223 $\pm$ 0.074 | 2.657 $\pm$ 0.030 |
| Mistral Large 2 | CoT + 3-Shot | 0.999 $\pm$ 0.016 | **0.032** $\pm$ 0.023 | 0.417 $\pm$ 0.154 |
| Mistral Large 2 | S2A | 1.018 $\pm$ 0.027 | 1.009 $\pm$ 0.129 | 0.454 $\pm$ 0.193 |
| Mistral Large 2 | S2A + CoT | 1.029 $\pm$ 0.025 | 0.663 $\pm$ 0.231 | 2.586 $\pm$ 0.132 |
| Mistral Large 2 | Social LM | 0.903 $\pm$ 0.020 | 2.021 $\pm$ 0.197 | 2.668 $\pm$ 0.021 |
| Mistral Large 2 | Plans/Insights* | 1.330 $\pm$ 0.021 | 0.089 $\pm$ 0.051 | 0.717 $\pm$ 0.140 |
| Mistral Large 2 | Reflexion 1 | 0.987 $\pm$ 0.016 | 2.311 $\pm$ 0.152 | 2.544 $\pm$ 0.049 |
| Mistral Large 2 | Reflexion 3 | 1.037 $\pm$ 0.020 | 1.772 $\pm$ 0.166 | 2.449 $\pm$ 0.065 |
| LLAMA-3 70B Instruct | QA | 1.174 $\pm$ 0.023 | 1.377 $\pm$ 0.110 | 2.770 $\pm$ 0.041 |
| LLAMA-3 70B Instruct | CoT | 1.029 $\pm$ 0.019 | 4.468 $\pm$ 0.176 | 2.761 $\pm$ 0.023 |
| LLAMA-3 70B Instruct | CoT + 3-Shot | 0.968 $\pm$ 0.020 | 3.065 $\pm$ 0.237 | 2.565 $\pm$ 0.066 |
| LLAMA-3 70B Instruct | S2A | 1.025 $\pm$ 0.026 | 2.100 $\pm$ 0.149 | 2.606 $\pm$ 0.060 |
| LLAMA-3 70B Instruct | S2A + CoT | 1.047 $\pm$ 0.048 | 2.938 $\pm$ 0.258 | 2.740 $\pm$ 0.031 |
| LLAMA-3 70B Instruct | Social LM | 1.112 $\pm$ 0.023 | 2.448 $\pm$ 0.335 | 2.759 $\pm$ 0.012 |
| LLAMA-3 70B Instruct | Plans/Insights* | 1.389 $\pm$ 0.051 | 3.127 $\pm$ 0.229 | 2.547 $\pm$ 0.056 |
| LLAMA-3 70B Instruct | Reflexion 1 | 0.953 $\pm$ 0.023 | 5.758 $\pm$ 0.152 | 2.57 $\pm$ 0.031 |
| LLAMA-3 70B Instruct | Reflexion 3 | 1.003 $\pm$ 0.016 | 5.531 $\pm$ 0.105 | 2.442 $\pm$ 0.029 |

Table 10: Comparing LLM models and prompting strategies when playing with tit for tat style strategies for the other agent across Rock, Paper, Scissors (RPS), Iterated Battle of Sexes (IBS), and the Iterated Prisoner's Dilemma (IPD).

IPD. Analogously, in RPS we always play the best response to the other agent's action at the last step and in IBS we always play the same action that the other agent did at the last step. We also add the very strong Mistral Large 2 model to match LLAMA-3 models but within the Mistral family. In most cases the LLMs perform quite poorly. All models are again much worse than the tabular model with the exception of CoT Prompting for Mistral Large 2 playing IBS. In this case, it even outperforms RMax with 3 examples, which makes sense given the payoff table it knows and the examples it is given which RMax does not receive. However, this model is inconsistent and, for example, does not perform well at RPS. Interestingly, CoT prompting seems to hurt the LLAMA-3 and Mixtral models. Social Prompting also doesn't seem to really provide benefits, which makes sense because the literal ToM performance falls off quite a bit in this setting.

## C.4. Additional Experiments on Action Space Inductive Bias

**The Good and Bad of Inductive Bias.** So far we have used the neutral action representations $J$, $F$, and $B$ to avoid contamination following prior work (Binz & Schulz, 2023; Akata et al., 2023). However, it is interesting given our results so far to get a better sense of the degree that prior knowledge in the LLMs impacts performance and the choice of action representation is great way for us to control the extent that this knowledge is evoked. For the best performing models LLAMA-3 70B Instruct and Mistral Large 2, we conduct experiments for all three games playing with both single action and tit for tat style partners. We provide comprehensive results for functional theory of mind performance in Figures 20, 22, 24, 26, 28, 30, 32, 34, 36, 38, 40, and 42, and for literal theory of mind performance in Figures 21, 23, 25, 27, 29, 31, 33, 35, 37, 39, 41, and 43. Here we compare neutral actions with the actual canonical action names for the game, the neutral actions

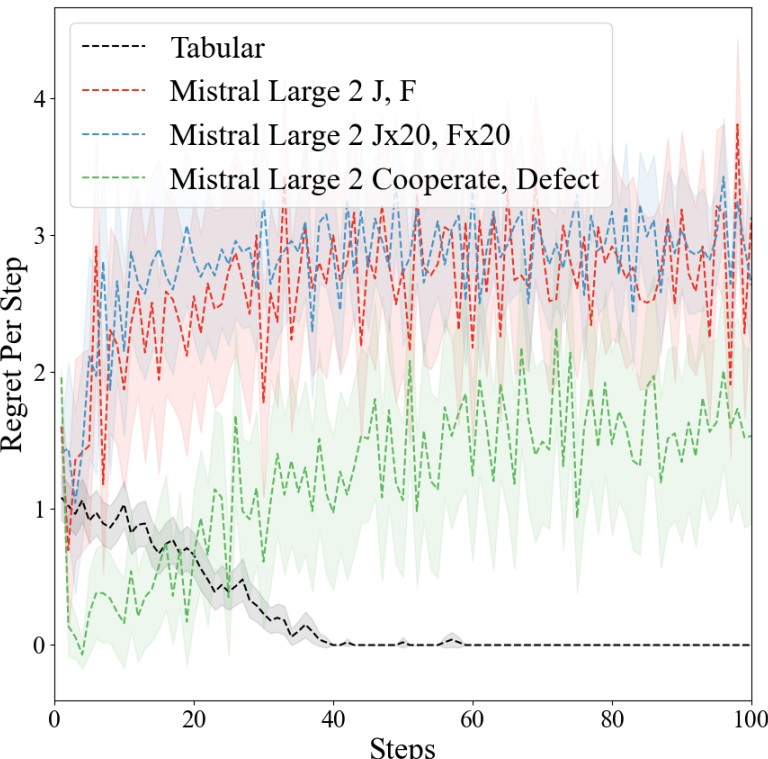

Figure 19: **IPD Regret for Mistral Large 2 with a Tit for Tat Partner Across Action Types**. We compare performance across action spaces with the Tabular RMax algorithm to test the influence of different levels of inductive bias.

with 20 repetitions to take up a longer portion of the context, and nonsense word actions (see Appendix C). We generally find that both LLM models experience systematic bias that prevents them from converging to optimal performance as the number of interactions grow. Indeed, only LLAMA-3 playing IBS with tit for tat partners (Figure 28) seems on the road to slow convergence. Meanwhile, literal theory of mind seems to be converging in a greater number of settings (Figures 29, 33, 35, and 37). We find that real actions often help with functional theory of mind performance early in the interaction stream while becoming harmful for converged performance as interaction goes on (Figures 19, 20, 26, 30, 32, and 36).

> **Impact of Inductive Bias**
>
> As exemplified by Figure 19, inductive bias can be helpful with a limited number of interactions while also hurting convergence in the long-term.

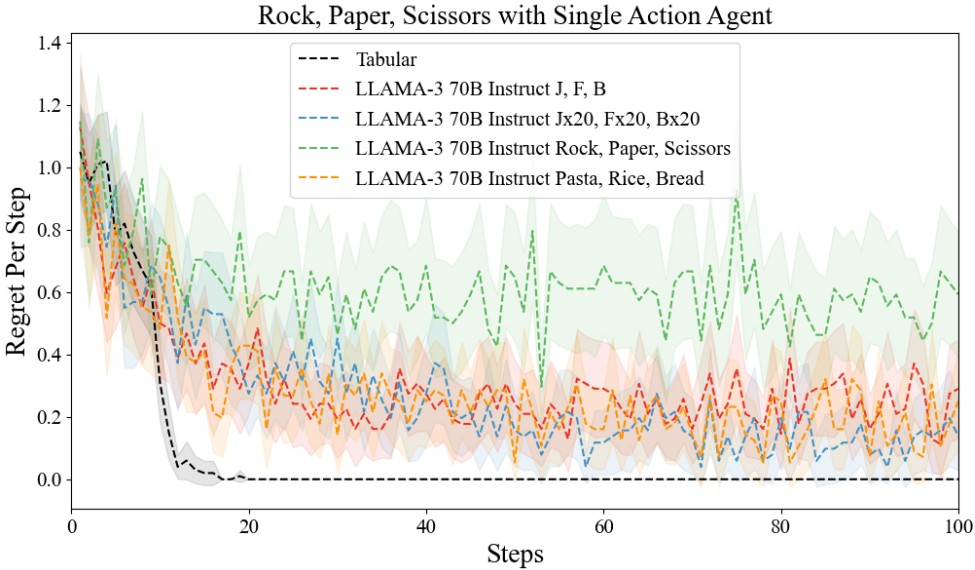

Figure 20: **RPS Regret for LLAMA-3 70B Instruct with a Single Action Partner Across Action Types**. We compare performance across action spaces with the Tabular RMax algorithm to test the influence of different levels of inductive bias.

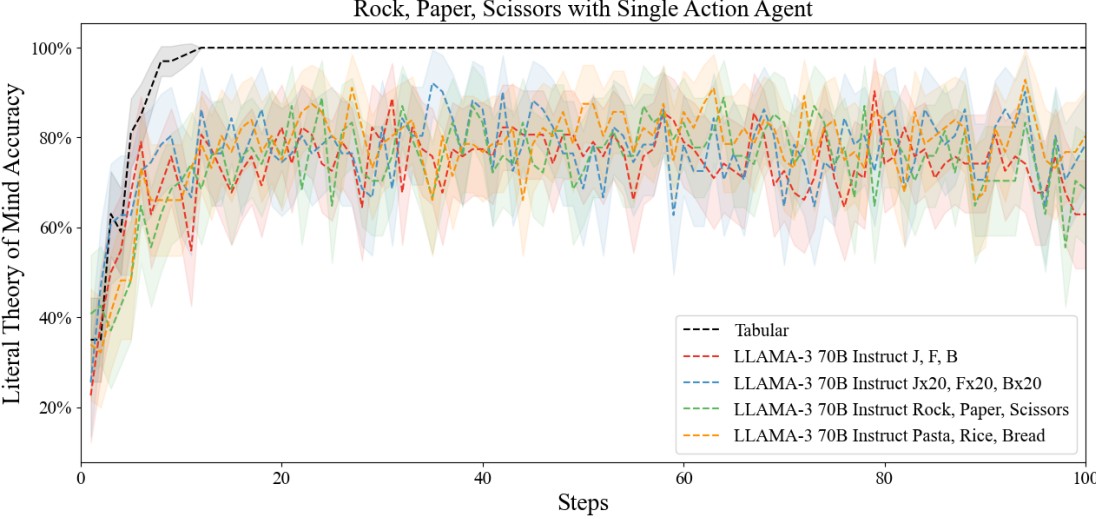

Figure 21: **RPS ToM % for LLAMA-3 70B Instruct with a Single Action Partner Across Action Types**. We compare accuracy across action spaces with a Tabular counting algorithm to test the influence of different levels of inductive bias.

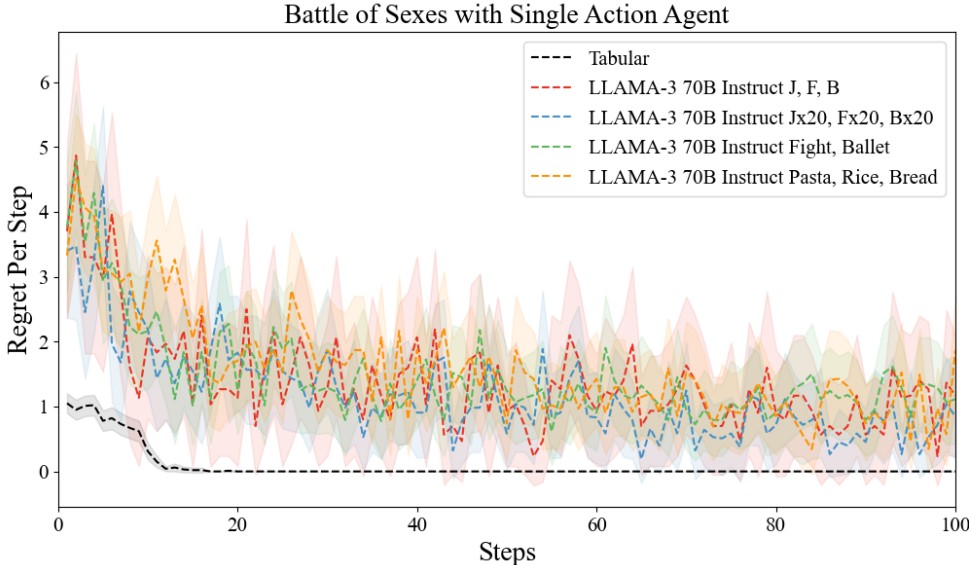

Figure 22: **IBS Regret for LLAMA-3 70B Instruct with a Single Action Partner Across Action Types**. We compare performance across action spaces with the Tabular RMax algorithm to test the influence of different levels of inductive bias.

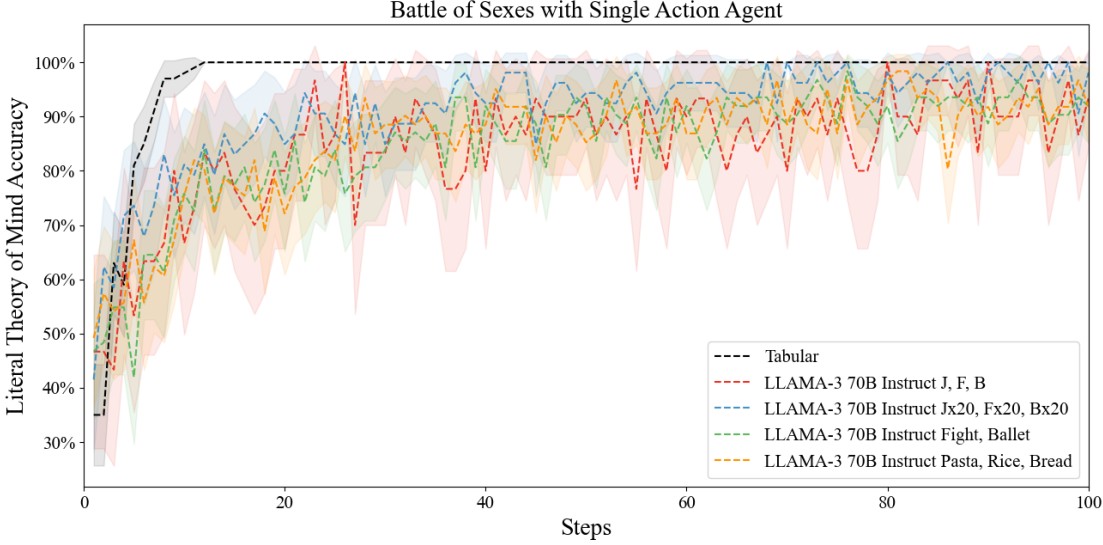

Figure 23: **IBS ToM % for LLAMA-3 70B Instruct with a Single Action Partner Across Action Types**. We compare accuracy across action spaces with a Tabular counting algorithm to test the influence of different levels of inductive bias.

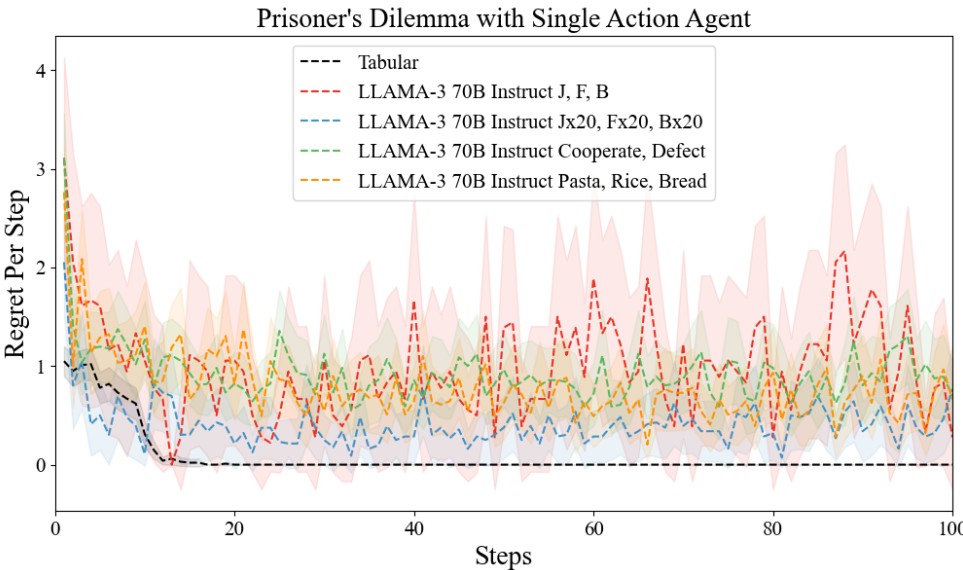

Figure 24: **IPD Regret for LLAMA-3 70B Instruct with a Single Action Partner Across Action Types**. We compare performance across action spaces with the Tabular RMax algorithm to test the influence of different levels of inductive bias.

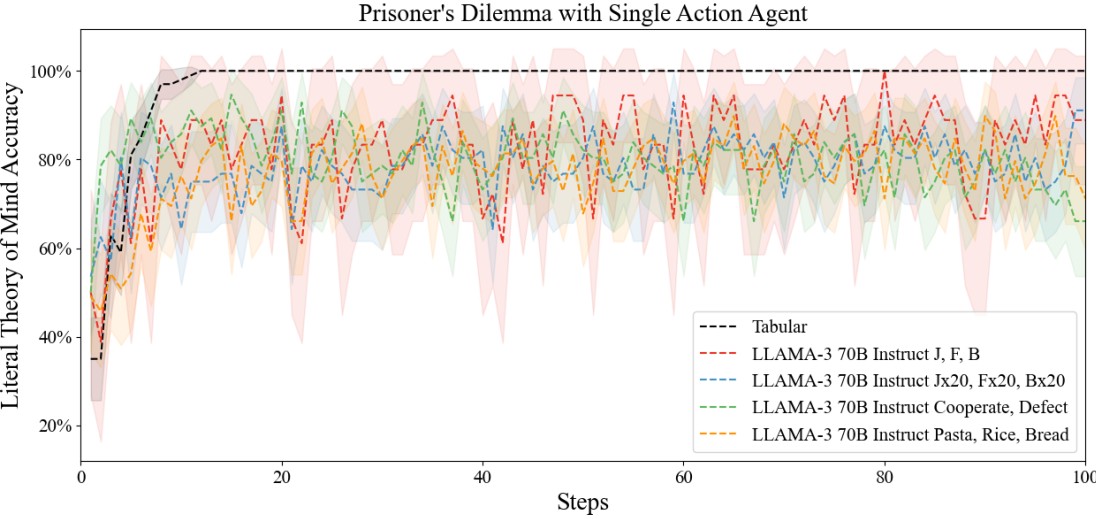

Figure 25: **IPD ToM % for LLAMA-3 70B Instruct with a Single Action Partner Across Action Types**. We compare accuracy across action spaces with a Tabular counting algorithm to test the influence of different levels of inductive bias.

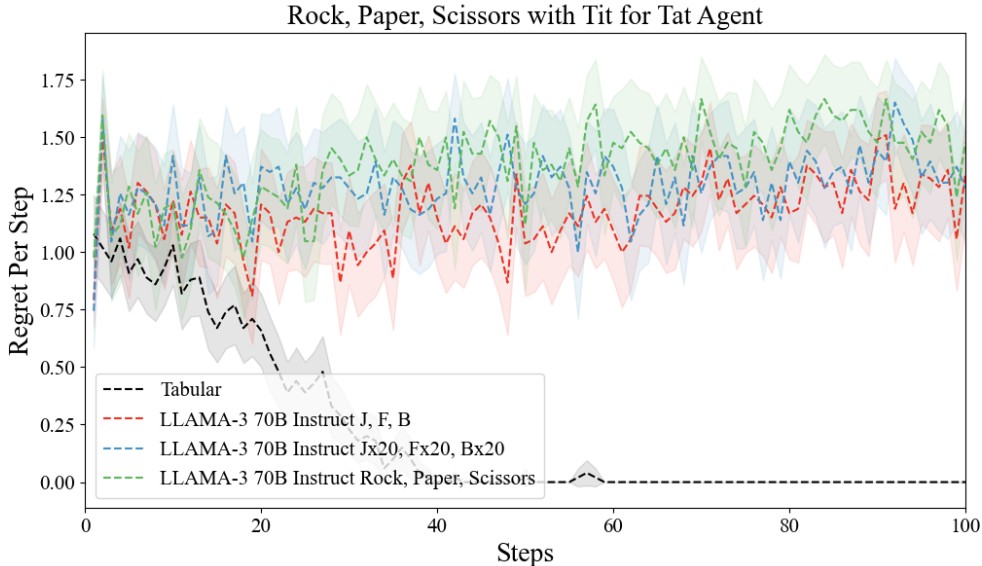

Figure 26: **RPS Regret for LLAMA-3 70B Instruct with a Tit for Tat Partner Across Action Types**. We compare performance across action spaces with the Tabular RMax algorithm to test the influence of different levels of inductive bias.

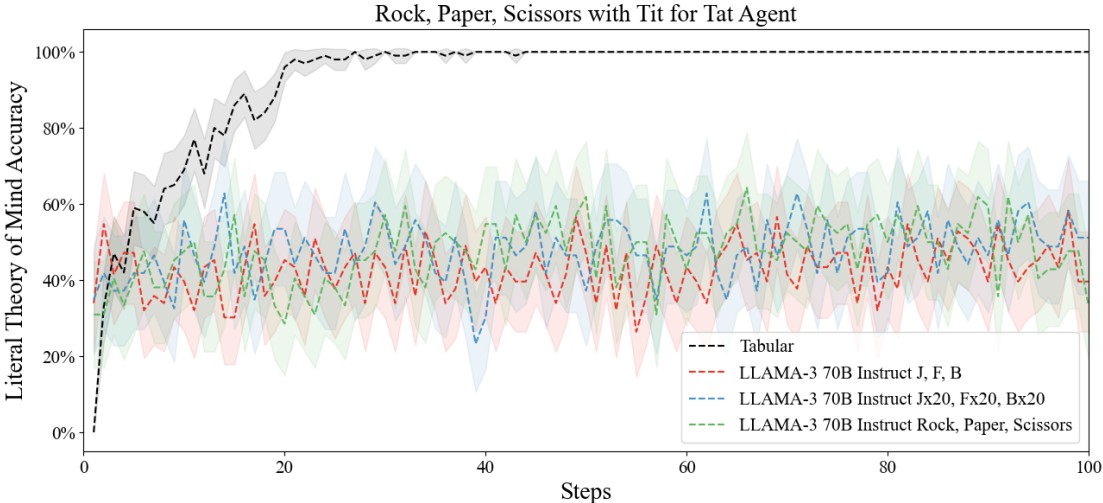

Figure 27: **RPS ToM % for LLAMA-3 70B Instruct with a Tit for Tat Partner Across Action Types**. We compare accuracy across action spaces with a Tabular counting algorithm to test the influence of different levels of inductive bias.

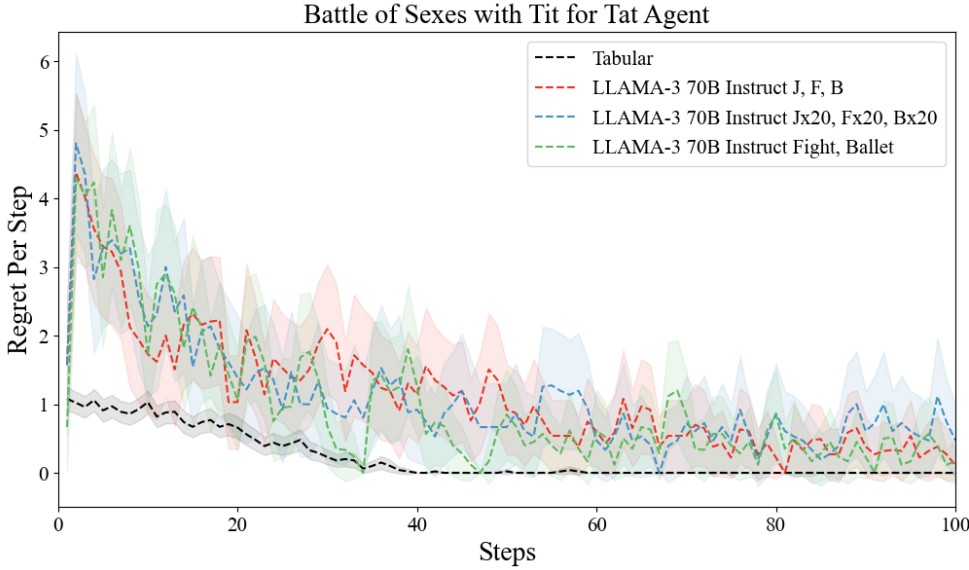

Figure 28: **IBS Regret for LLAMA-3 70B Instruct with a Tit for Tat Partner Across Action Types**. We compare performance across action spaces with the Tabular RMax algorithm to test the influence of different levels of inductive bias.

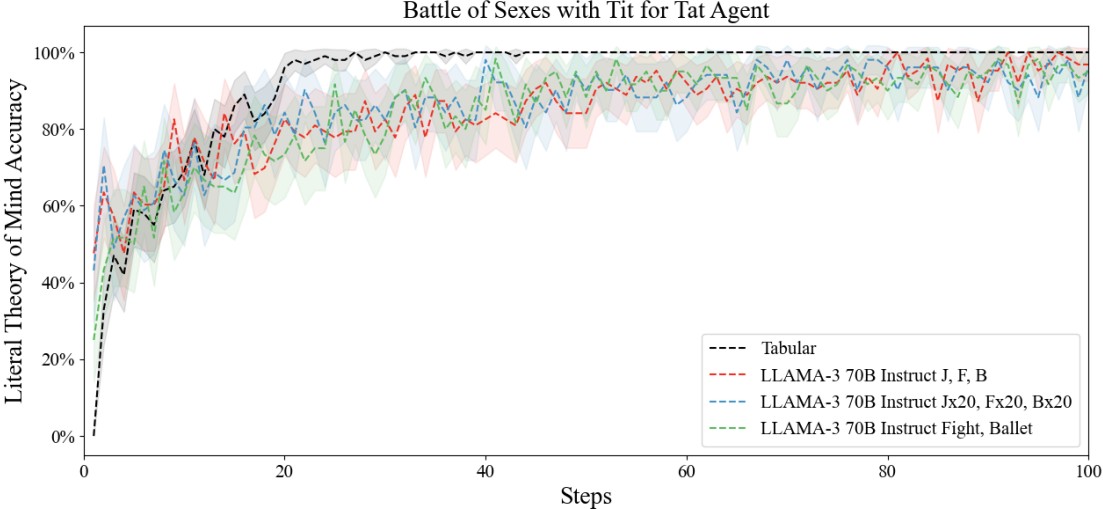

Figure 29: **IBS ToM % for LLAMA-3 70B Instruct with a Tit for Tat Partner Across Action Types**. We compare accuracy across action spaces with a Tabular counting algorithm to test the influence of different levels of inductive bias.

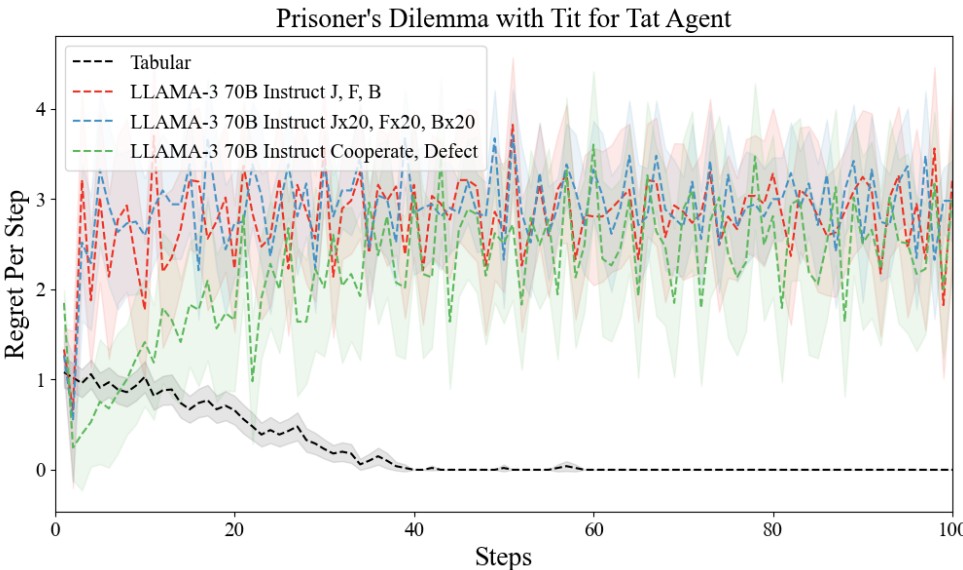

Figure 30: **IPD Regret for LLAMA-3 70B Instruct with a Tit for Tat Partner Across Action Types**. We compare performance across action spaces with the Tabular RMax algorithm to test the influence of different levels of inductive bias.

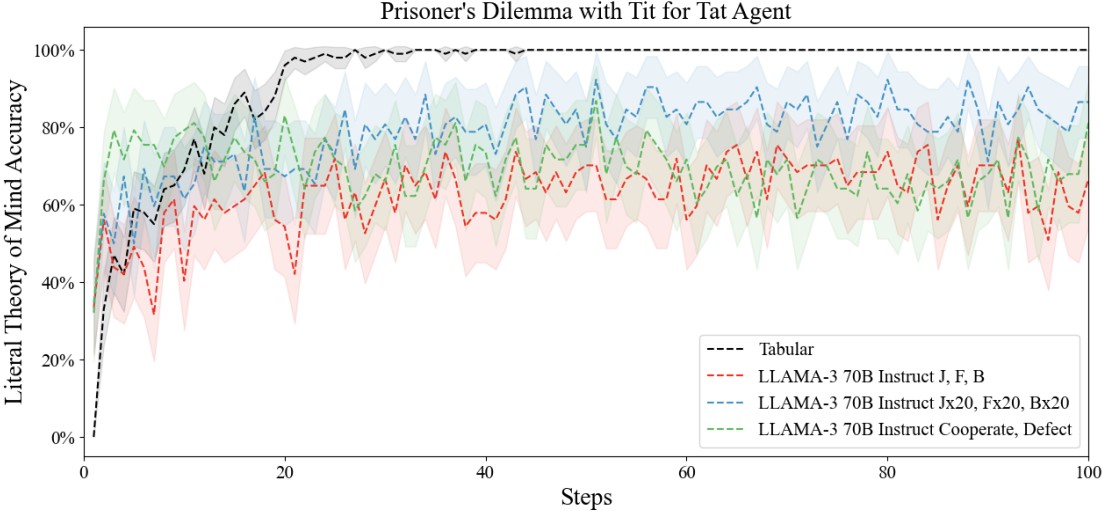

Figure 31: **IPD ToM % for LLAMA-3 70B Instruct with a Tit for Tat Partner Across Action Types**. We compare accuracy across action spaces with a Tabular counting algorithm to test the influence of different levels of inductive bias.

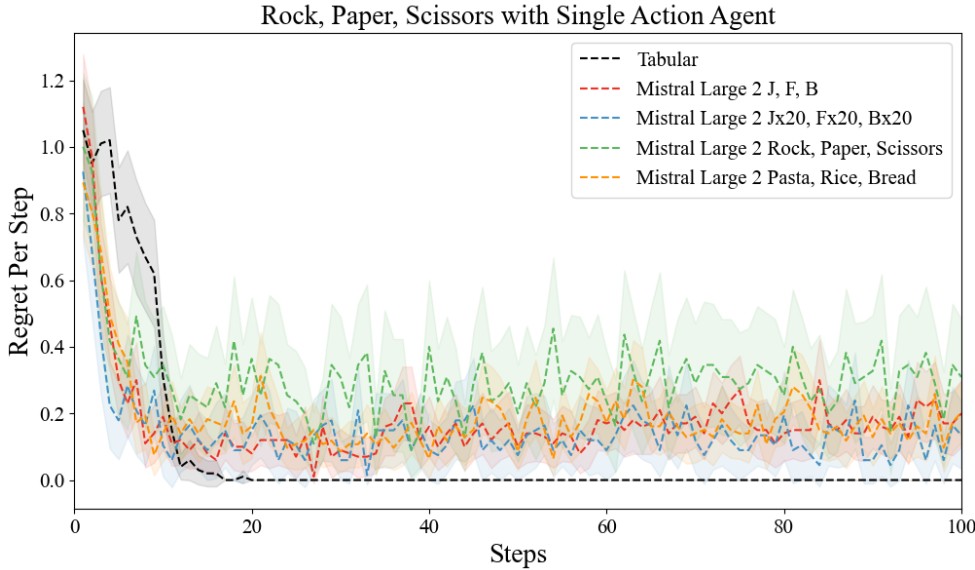

Figure 32: **RPS Regret for Mistral Large 2 with a Single Action Partner Across Action Types**. We compare performance across action spaces with the Tabular RMax algorithm to test the influence of different levels of inductive bias.

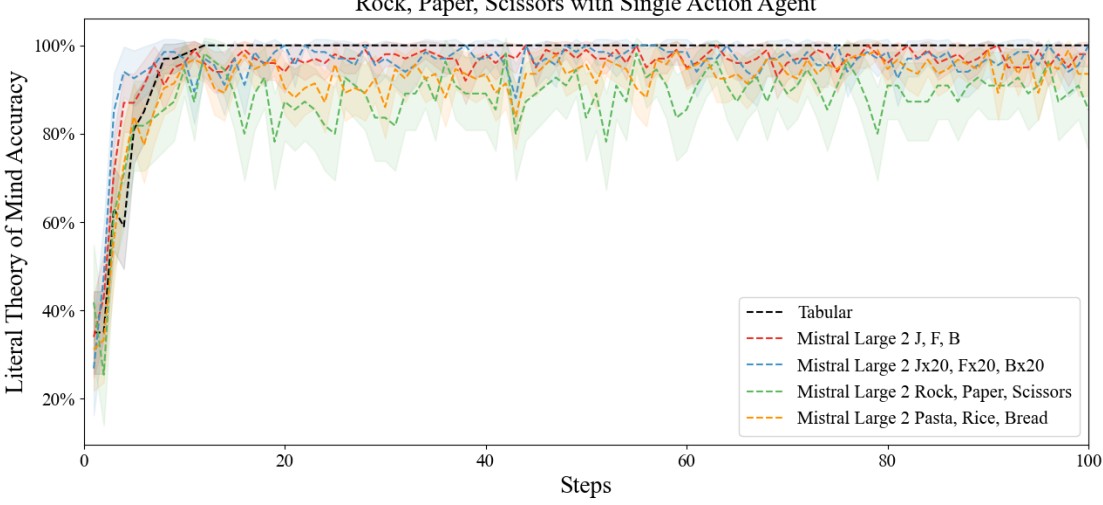

Figure 33: **RPS ToM % for Mistral Large 2 with a Single Action Partner Across Action Types**. We compare accuracy across action spaces with a Tabular counting algorithm to test the influence of different levels of inductive bias.

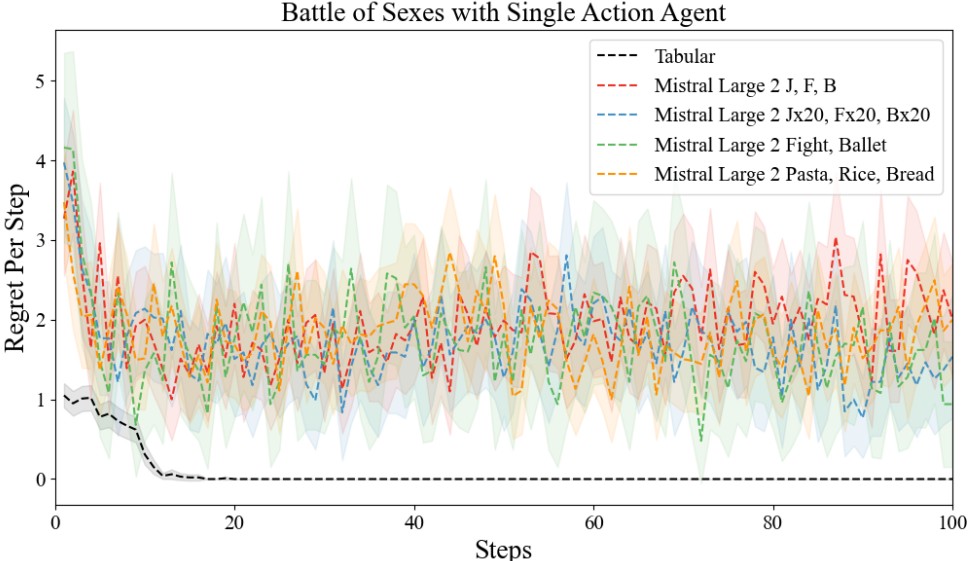

Figure 34: **IBS Regret for Mistral Large 2 with a Single Action Partner Across Action Types**. We compare performance across action spaces with the Tabular RMax algorithm to test the influence of different levels of inductive bias.

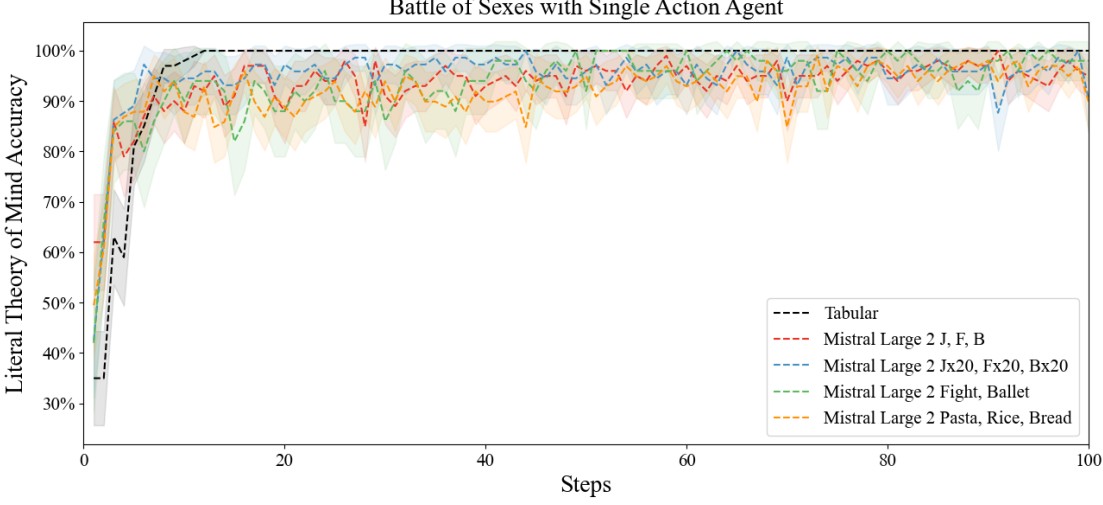

Figure 35: **IBS ToM % for Mistral Large 2 with a Single Action Partner Across Action Types**. We compare accuracy across action spaces with a Tabular counting algorithm to test the influence of different levels of inductive bias.

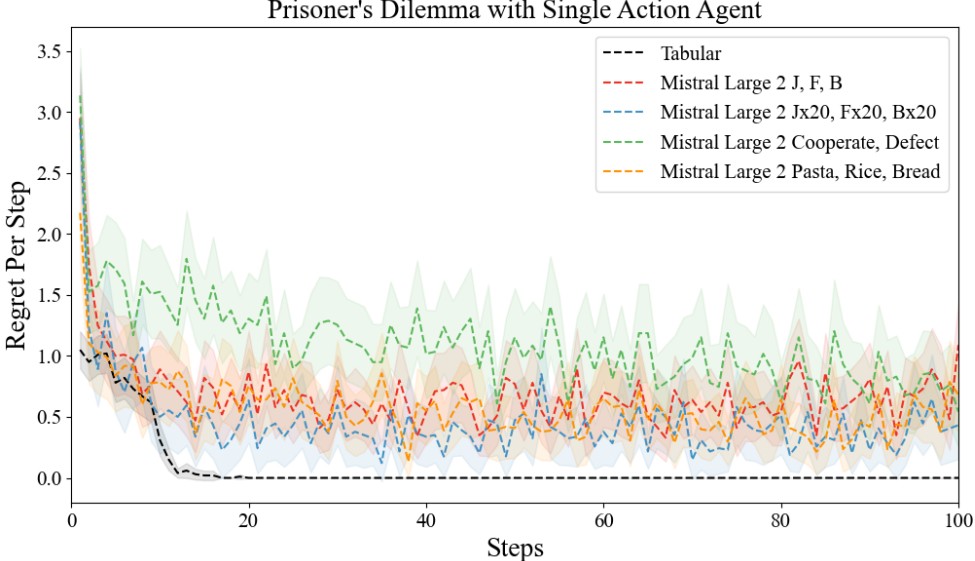

Figure 36: **IPD Regret for Mistral Large 2 with a Single Action Partner Across Action Types**. We compare performance across action spaces with the Tabular RMax algorithm to test the influence of different levels of inductive bias.

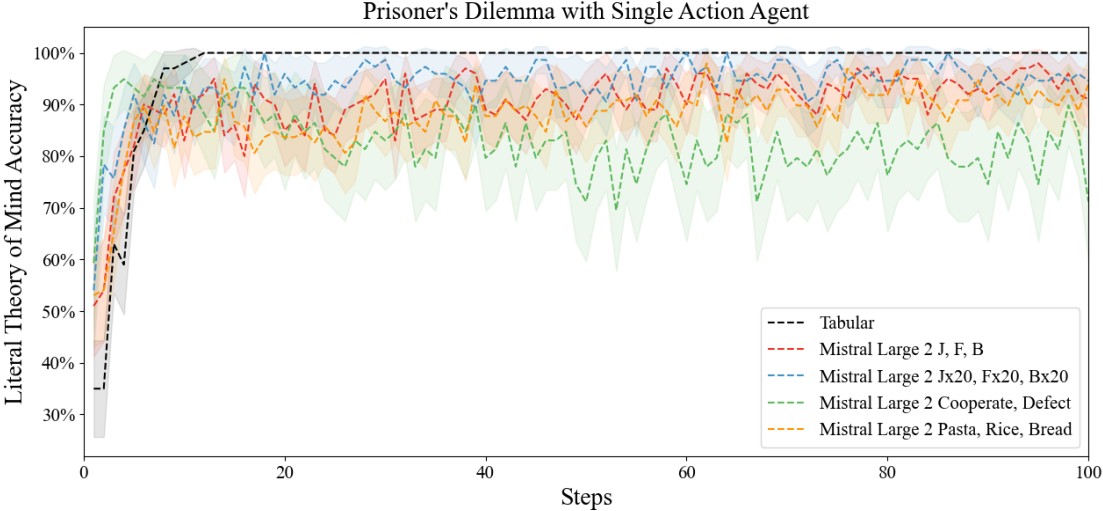

Figure 37: **IPD ToM % for Mistral Large 2 with a Single Action Partner Across Action Types**. We compare accuracy across action spaces with a Tabular counting algorithm to test the influence of different levels of inductive bias.

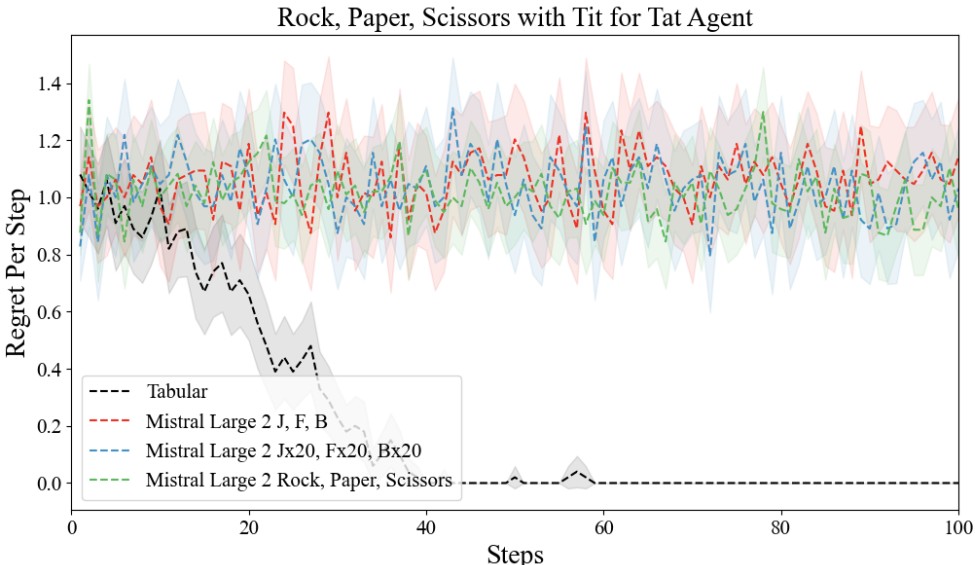

Figure 38: **RPS Regret for Mistral Large 2 with a Tit for Tat Partner Across Action Types**. We compare performance across action spaces with the Tabular RMax algorithm to test the influence of different levels of inductive bias.

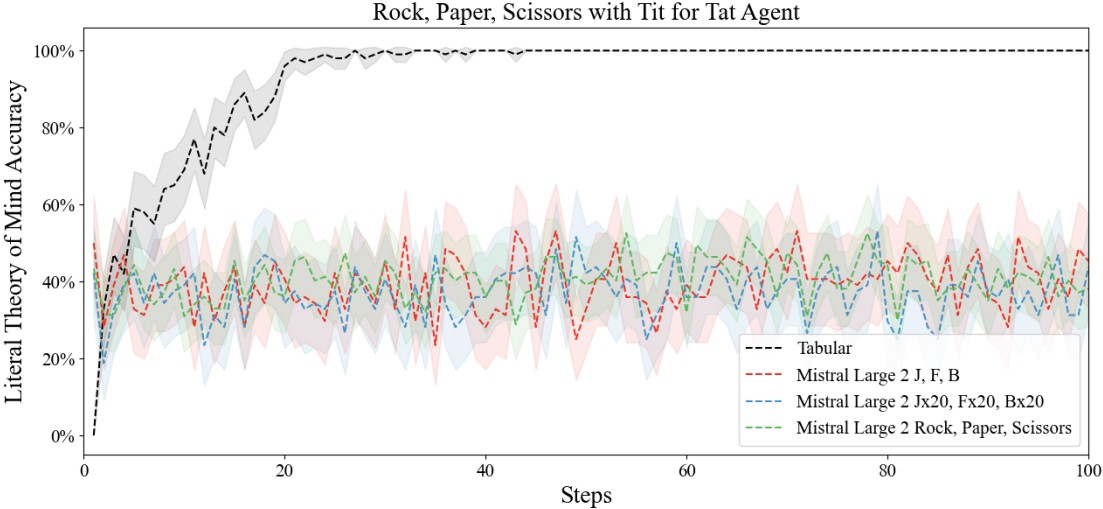

Figure 39: **RPS ToM % for Mistral Large 2 with a Tit for Tat Partner Across Action Types**. We compare accuracy across action spaces with a Tabular counting algorithm to test the influence of different levels of inductive bias.

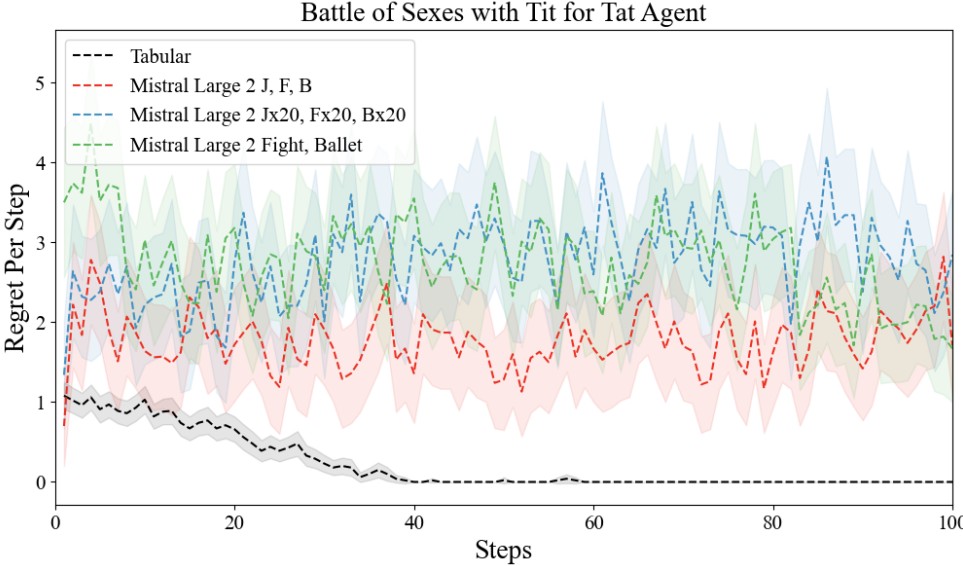

Figure 40: **IBS Regret for Mistral Large 2 with a Tit for Tat Partner Across Action Types**. We compare performance across action spaces with the Tabular RMax algorithm to test the influence of different levels of inductive bias.

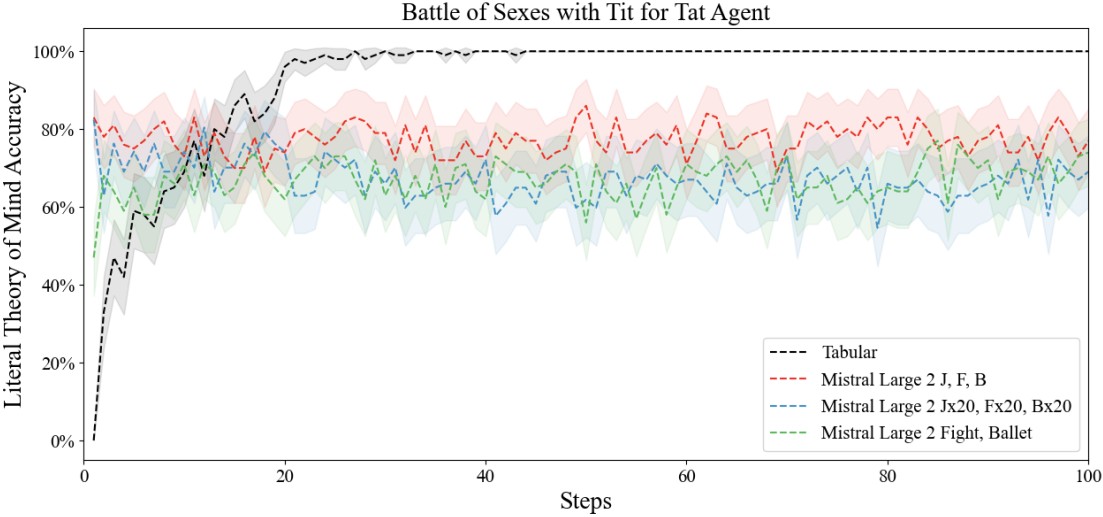

Figure 41: **IBS ToM % for Mistral Large 2 with a Tit for Tat Partner Across Action Types**. We compare accuracy across action spaces with a Tabular counting algorithm to test the influence of different levels of inductive bias.

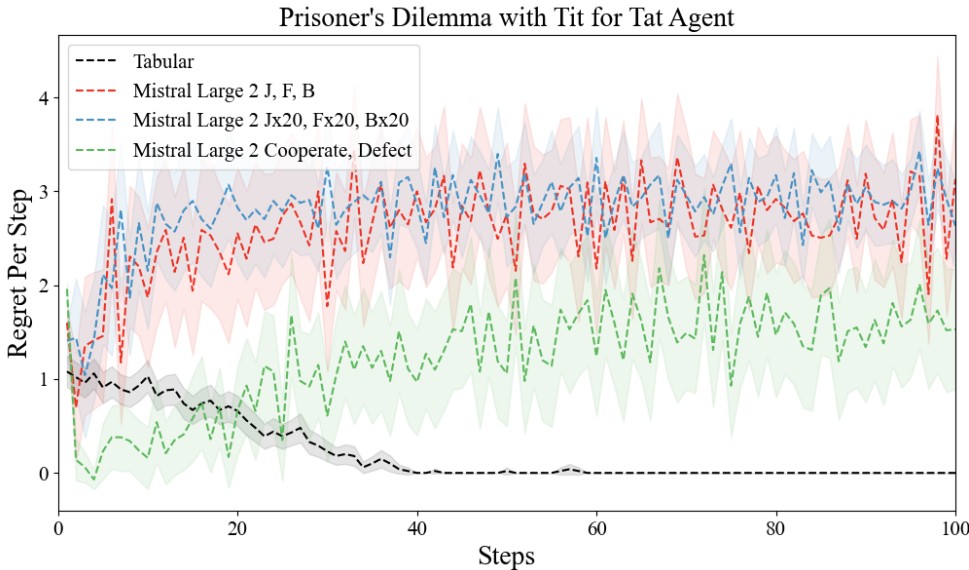

Figure 42: **IPD Regret for Mistral Large 2 with a Tit for Tat Partner Across Action Types**. We compare performance across action spaces with the Tabular RMax algorithm to test the influence of different levels of inductive bias.

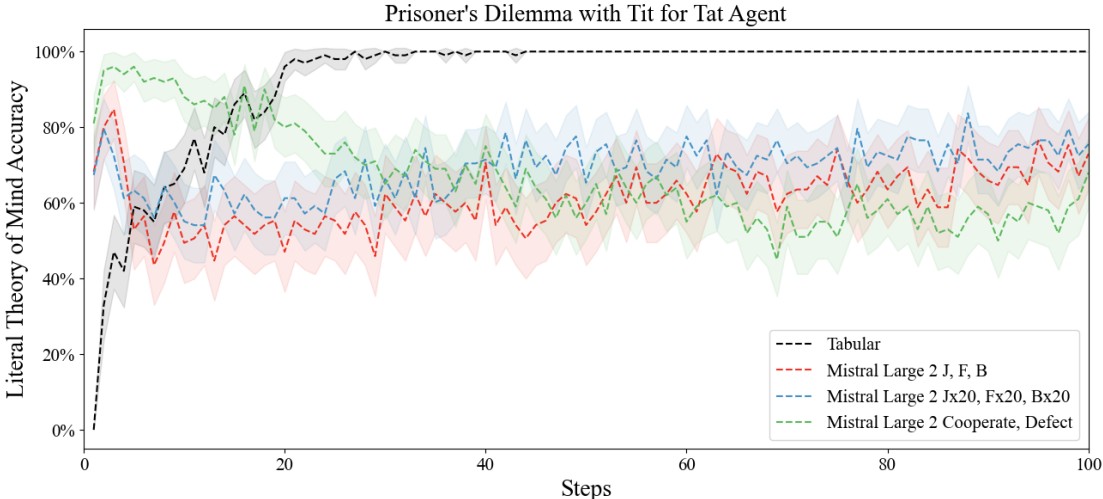

Figure 43: **IPD ToM % for Mistral Large 2 with a Tit for Tat Partner Across Action Types**. We compare accuracy across action spaces with a Tabular counting algorithm to test the influence of different levels of inductive bias.

