# OpenReview forum: "Position: Theory of Mind Benchmarks are Broken for Large Language Models"
_ICML.cc/2025/Position_Paper_Track — ICML 2025 Position Paper Track poster_

### Official Review · Reviewer_1Qq8 · 2025-03-06

**Significance:** 2
**Argument Clarity:** 3
**Rating:** 3
**Confidence:** 3

**Questions:**

The empirical evaluation focuses primarily on open-source models (LLAMA, Mixtral, Mistral) without including models known for stronger reasoning capabilities. Could the author extend to models that are better at reasoning like Chatgpt or Claude?

**Discussion Potential:**

3

**Paper Summary:**

This paper raises an important issue regarding the evaluation of theory of mind (ToM) capabilities in LLMs, particularly the distinction between literal theory of mind (predicting behavior) and functional theory of mind (acting rationally based on those predictions). The authors argue that current benchmarks fail to capture functional ToM, leading to misleading conclusions about LLMs’ true capabilities. The experiments with repeated matrix games provide empirical support, showing that LLMs may correctly predict an opponent’s strategy but still fail to act optimally in response.


## update after rebuttal
The rebuttal addresses some of my concerns and I have increased my score.

**Position:**

Yes

**Position In Title:**

Yes

**Related Work:**

3

**Strengths And Weaknesses:**

Strengths:

Clear Position Statement: The paper clearly articulates its position that "current theory of mind benchmarks are broken because they only measure literal theory of mind rather than functional theory of mind." This stance is consistently referenced throughout the paper. Empirical Evidence: The authors provide substantial empirical evidence through experiments across multiple games (RPS, IBS, IPD) and multiple models to support their position. The inclusion of various prompting strategies strengthens their case. Formal Definitions: The paper offers formal definitions (2.1 and 2.2) that clearly delineate the two types of theory of mind, making their position precise and testable. Alternative Views: The paper explicitly addresses alternative perspectives, including the game-theoretic view and the potential sufficiency of literal theory of mind for certain applications, fulfilling the requirement to engage with opposing viewpoints.


Weaknesses:

Potential Conflation with Reasoning: The paper may conflate functional theory of mind with general reasoning capabilities. The observed limitations could be due to reasoning deficiencies rather than specifically theory of mind issues, making the framing potentially misleading.

Connection with predict and optimize and predict then optimize:

I feel this paper is also related to end-to-end predict-and-optimize and 2-stage predict-then-optimize literature [1]. In the end-to-end predict-and-optimize literature, it has been demonstrated that end-to-end approaches that jointly optimize prediction and decision-making often outperform two-stage approaches where prediction is performed independently from optimization. The primary reason is that not all prediction errors are equally consequential for the final decision quality. End-to-end approaches can learn to focus prediction accuracy on the parameters that most significantly impact the objective function, rather than treating all prediction targets as equally important. This directly relates to the paper's distinction between literal and functional theory of mind. The authors could strengthen their argument by framing their findings within this established paradigm: current theory of mind benchmarks essentially evaluate only the "predict" component of a "predict-then-optimize" pipeline, ignoring how prediction errors propagate to decision quality. The paper would benefit from a concrete example demonstrating how different types of prediction errors in theory of mind tasks have varying impacts on decision quality. For instance, in their game theory experiments, misclassifying an opponent's strategy in certain ways might be much more detrimental to performance than others. Then it could strengthen the view of "broken". Currently, based on the experiments, LLMs tested are good at prediction, bad at optimization, bad at two-stage predict then optimize and bad at end-to-end predict-and-optimize. It is hard to distinguish if there is a benefit for predict-and-optimize together.

[1] Elmachtoub A N, Grigas P. Smart “predict, then optimize”[J]. Management Science, 2022, 68(1): 9-26.

**Support:**

2

---

> ### Author Rebuttal · Authors · 2025-04-01
>
> Thank you for your insightful review of our paper. You have shared some really thought provoking comments and suggestions. Addressing your concerns will greatly improve our paper for the final draft.
>
> **Concern “Potential Conflation with Reasoning”:** We are very sorry to hear that you found our framing confusing or misleading. As pointed out by Reviewer sQuU, part of the difficulty here is that the term “theory of mind” is used in slightly different ways in different bodies of literature and even in industry.  We will revise the paper to be much clearer about exactly what we mean and don’t mean by theory of mind. We are using the term “theory of mind” as an idiomatic description of behavior whereas it can also be interpreted more literally to mean there is an actual artifact called a "theory of mind" which is an explicit description or model of an agent's thinking. As you pointed out, literal theory of mind can be alternatively called theory of mind prediction. But it is worth mentioning that even literal theory of mind could potentially still be achieved to optimal precision without forming an explicit description of an agent’s thinking. For example, the model may be conditioning on a variable that is perfectly correlated with the variables another agent is actually considering. The literature on theory of mind in machine learning is generally unconcerned about this distinction. If this prediction ability is achieved, it is not of practical consequence if there actually is a theory of mind model. Likewise, functional theory of mind can be alternatively called theory of mind reasoning (or “optimization” as you pointed out). Our assertion in this paper is that functional theory of mind is typically what people actually care about when they say that they want to endow LLM agents with better theory of mind capabilities. Notice that this goal requires the ability to adapt to other agents, but does not necessarily require that the agent has an explicit theory of mind model. Agents may have to reason about the minds of other agents during this adaptation process, but it is not possible to totally disentangle this reasoning from the capability of theory of mind itself. If we are describing a capability, the RL perspective is to focus on the problem it solves, which must be disentangled from the solution method. To define the capability of theory of mind in such a way that an explicit theory of mind model is needed would only make sense if there were kinds of solutions expressible with model-based approaches that are not expressible with model-free approaches. However, we know that these algorithm classes have the same expressiveness.
>
> **Connections to predict-then-optimize framework:** Thank you for bringing up this very relevant conceptual framework that we were largely unaware of at the time of our submission. Will add a paragraph on predict-then-optimize to section 3. This is indeed important related work and motivation. We will also add tables in the appendix, complimenting the payoff tables, that highlight the impact of particular action misclassifications on the performance in each game as you suggested. Although we are still not sure whether we interpreted your comment correctly that “it is hard to distinguish if there is a benefit for predict-and-optimize together.” We think that this may reflect a misunderstanding of the point of our paper i.e. that we are trying to show that literal theory of mind performance is needed for functional theory of mind performance. Hopefully we have addressed this confusion above.
>
> **Question “models that are better at reasoning”:** We have added comprehensive experiments with the open source “DeepSeek-R1-Distill-Qwen-32B" model that is known for superior reasoning. It was released 3 days prior to the abstract deadline. Overall, we find that the model has the best functional theory of mind of any model we tried while quite interestingly displaying far worse literal theory of mind performance. This once again serves as validation that these LLMs lack process consistency and that strong function theory of mind performance and strong literal theory of mind performance do not imply each other. For example, when playing Rock, Paper, Scissors with the canonical action space, the functional theory of mind regret is the best we have seen from LLM models and comparable to that of the tabular model. However, the literal theory of mind performance is merely 63.7% (far worse than any other LLM in Table 1). It is a very fascinating result as it acts like a rational LLM with a literal theory of mind performance nearing 98% yet somehow it fails to predict the other agents actions when it is actually asked to. Please let us know if you believe these results are sufficient. Using the token lengths from our DeepSeek experiments to provide an estimate, it would cost ~1k USD to run OpenAI o3-mini across our experiments, and ~13.7k USD to run OpenAI o1 across our experiments.

---

### Official Review · Reviewer_Pqm1 · 2025-03-13

**Significance:** 2
**Argument Clarity:** 3
**Rating:** 3
**Confidence:** 4

**Questions:**

N/A

**Discussion Potential:**

2

**Paper Summary:**

This paper claims that current theory of mind (ToM) reasoning in LLMs are broken. To support their argument, the authors gives an example  playing the game of Rock, Scissors, and Paper against an agent.

### Update after rebuttal
The authors have clarified my concerns. Hopefully, the authors will improve their writing and address the recent AAAI paper.

The paper is really interesting and has potential to foster discussion.

I have raised my score.

**Position:**

Yes

**Position In Title:**

Yes

**Related Work:**

3

**Strengths And Weaknesses:**

Strengths
- The topic will likely foster discussions in ICML community
- Example of integrating ToM in an agent-based game
- Discusses alternative view


Weakness
- The major concern of the definition 2.1 and 2.2 that  the authors formulated any reinforcement learning based algorithm in can be formulated as as ToM task. Whereas, ToM in reality has two main criteria: i) dynamically updates on belief based on others belief and actions and ii) tries to understand other persons perspective.

The authors' formulation supports criteria (i) but ignores the second one.

An example of how ToM can be integrated into a game, Codenames, can be the following paper [1](available on arXiv since December 2024 and accepted to AAAI 2025). The authors provide "clues" as a ToM task to the guesser with the hypothesis that the guesser has a belief state over all possible states of the world (board card assignments).

References
1. Improving Cooperation in Language Games with Bayesian Inference and the Cognitive Hierarchy. Joseph Bills, Diego Blaylock, Christopher Archibald.

**Support:**

2

---

> ### Author Rebuttal · Authors · 2025-04-01
>
> We really appreciate your comment suggesting that our paper will likely foster discussion within the community. This was a key goal of ours in writing this position paper. On the other hand, we were disheartened to see your low overall impression of our paper. We believe that this may stem from some misunderstandings regarding our submission.
>
> **Concern “any RL based algorithm in can be formulated as a ToM task”:** Thank you for making this point as it gave us a lot more insight into why your perspective on our paper may be so negative. As this is a key concern of yours about our paper, we have broken our attempt to address it into two parts.
>
> **Part 1: This was not the goal of our submission.** We should start by saying that we feel that being overly focused on this point is slightly misunderstanding the goals of our paper. We were not even attempting with these metrics to characterize the set of tasks that are interesting for theory of mind. We feel the community already has a good handle on this. What we were instead commenting on is the method of evaluating the theory of mind capabilities of agents on these tasks where we know it is highlighted. Thank you for mentioning the very enjoyable game Codenames. It is a great example for highlighting our point. In Codenames, literal theory of mind could naturally be measured in terms of guesses by the guesser about the board-card assignments. These assignment can be retrieved from some transformation of the actions that the clue giver would take in different game states following Definition 2.1.  Functional theory of mind would be the guesser’s regret with respect to the actual team performance on the game following Definition 2.2, which also includes an aspect of reasoning on top of a literal theory of mind model. For example, given the context of the game, guessers may decide to save certain guesses for later to avoid the potential negative consequences of guessing wrong and potentially gifting the other team an opportunity. We argue in this paper, that it is quite possible that a Codenames agent could perform well in terms of literal theory of mind on this kind of theory of mind task while simultaneously struggling to turn that into a rational strategy, displaying poor functional theory of mind as a result.
>
> **Part 2: We can add a metric to measure how interesting a task is for theory of mind.** While this was not our initial goal, we agree that it can be an interesting contribution as well. As such, building off Definition 2.2, we will provide a new Definition 2.3 measuring how interesting a task is for theory of mind in the revised draft. The task must include a distribution over the joint policies of the other agents $\pi^{-i}$ and our metric would be an expectation over two joint policies $\pi’^{-i}$ and $\pi’’^{-i}$ drawn from this distribution. We can then define $\pi’^{i^*}$ as the optimal response to $\pi’^{-i}$. The metric would be the expected functional theory of mind performance i.e. regret of $\pi’^{i^*}$ in the context of $\pi’’^{-i}$. This metric requires that the task itself highlights the need to adapt to the specific policies of the other agents. Hopefully this also makes it clearer how Definition 2.2 relates to your proposed criteria (ii) about understanding the perspectives of other agents.
>
> **Concern about connection to Codenames:** We also really appreciate you mentioning Codenames because it highlights a subtlety in our argument. In games like Codenames, because of the focus on multi-agent interaction, it is already natural to consider the performance of agents in terms of functional theory of mind i.e. regret or returns. Our argument is definitely not that benchmarks like Codenames are broken, but rather that evaluation suites that focus on false belief tasks and don’t include interactive scenarios like Codenames can be misleading. We will add a section to the end of the paper with our extra space in the final draft that discusses what kind of benchmarks we are advocating for. In this section we will be sure to highlight that games like Codenames, Hanabi, Taboo, and Wavelength are exactly the kind of evaluation we are advocating should be included in evaluation suites for theory of mind in LLMs.
>
> **Concern “The authors' formulation supports criteria (i) but ignores (ii)” -- trying to understand the perspective of others:** We believe that this point that you made is factually inaccurate and that your confusion about this point may be a major reason for your negative impression of our paper. A subtlety that could have been missed regarding Definition 2.2 is that it is conditional on the joint policy of the other agents $\pi^{-i}$. We tried to highlight this in the text, but will revise the math to make the conditional dependence more clear. This is very much that case in our experiments, for example, an agent may perform arbitrarily poorly on Rock, Paper, Scissors if they do not adapt their policy to the policy of the other agent.

---

> > ### Comment · Reviewer_Pqm1 · 2025-04-05
> >
> > Thank you authors for clarifying my concerns. The writing was not clear as I raised my concerns. Hopefully, the authors will improve their writing and address the recent AAAI paper.
> >
> > The paper is really interesting and has potential to foster discussion.
> >
> > I have raised my score.

---

> > > ### Author Response · Authors · 2025-04-07
> > >
> > > Thank you for your comment. We really appreciate your support of our paper. Your review has been very helpful in identifying areas where we can improve the writing to better communicate our main ideas. We have already begun revising the text to address these points of confusion. Fortunately, the additional space available for the camera-ready version will also allow us to prominently highlight the connection with the AAAI paper on Codenames and related theory of mind games.

---

### Official Review · Reviewer_sm5Y · 2025-03-13

**Significance:** 3
**Argument Clarity:** 3
**Rating:** 3
**Confidence:** 3

**Questions:**

1. Given the simplification concern, have the authors considered testing their hypothesis in more complex, realistic multi-agent scenarios beyond classical game theory benchmarks? Have the authors considered how results might change if human participants, or dynamically learning AI agents, were included in the interaction loop, introducing additional variations in policies and behaviors?
2. Specifically, for multi-agent collaboration or adversarial competition, do the authors have proposals for new benchmarks or performance metrics that capture both literal and functional ToM more realistically?
3. Can the authors provide more insights or hypothethis regarding why Social Prompting strategies still show considerable gaps despite explicitly providing theory-of-mind predictions as inputs?

**Discussion Potential:**

3

**Paper Summary:**

This paper argues that current benchmarks assessing Theory of Mind capabilities in LLMs are inadequate because they primarily measure "literal theory of mind"—the ability to predict other agents’ actions—instead of "functional theory of mind," defined as adapting rationally in-context based on these predictions. The authors argue that strong literal ToM performance doesn't imply robust functional ToM performance. Through experiments involving simple games like Rock, Paper, Scissors, Iterated Battle of Sexes, and Iterated Prisoner's Dilemma, they illustrate that even top-performing open-source models struggle significantly with functional ToM despite strong literal ToM capabilities.

**Position:**

Yes

**Position In Title:**

Yes

**Related Work:**

3

**Strengths And Weaknesses:**

Strengths:
- The paper clearly delineates literal from functional theory of mind, highlighting a critical and often overlooked distinction.
- The provided empirical examples effectively illustrate the authors' points, making the discussion concrete.
- The topic is highly relevant in the LLM evaluation community, as well as the trustworthiness and explanability field.

Weaknesses:
- The experiments primarily focus on static or one-shot opponents (e.g., always choosing “Rock” in RPS). Real-world interactions typically involve multiple agents simultaneously learning and adapting to each other. Hence, results from such simplified interactions might not fully generalize to the complexities of real-life multi-agent scenarios.
- Following the above point, although the paper’s experiments on Rock, Paper, Scissors, Iterated Battle of Sexes, and Iterated Prisoner’s Dilemma are illustrative, they are relatively short-horizon or involve static partners. It remains unclear how the findings extend to larger-scale, more dynamic multi-agent collaborations or competitive settings with longer or more varied interaction histories.

Typo: Line 356 strugle->struggle

**Support:**

3

---

> ### Author Rebuttal · Authors · 2025-04-01
>
> Thank you for your review. It provided us with some very insightful feedback that will improve our paper after subsequent updates. Please see our attempt to address your key concerns and questions below:
>
> **Concern “experiments primarily focus on static or one-shot opponents”:**  We would like to emphasize that in the submitted draft we did provide a comprehensive set of experiments with tit-for-tat partners. These strategies are generally considered dynamic and somewhat human-like (Akata et al., 2023). Due to space constraints, we were only able to highlight this in the paragraph titled “Deeper Analysis” at the end of section 4. However, we did provide detailed results in Table 9, Figure 19, Figures 26-31, and Figures 38-43 of the appendix. Luckily, we will be given an extra page for the final draft and could include much more discussion of these results with that space. Overall, we find, in this more complex scenario, that both literal theory of mind and functional theory of mind are harder to achieve. Importantly, they do not necessarily go hand in hand. For example, in Figure 30 all LLMs experience much worse performance with respect to functional theory of mind than would be implied by their literal theory of mind performance in Figure 31.
>
> **Concern “results from such simplified interactions might not generalize”:** It is interesting for us to see this concern mentioned in our review. It is really very much opposite to the way we had been thinking about it. We had actually seen the simplified nature of the tasks in our paper as one of its biggest selling points. In complex tasks, there are generally many more considerations (as you mentioned) other than theory of mind, and it is harder to pin down exactly why something works or doesn’t without introducing conflating factors. We don’t quite understand why we would have confidence that agents that can’t perform simple tasks would be able to do more complex tasks. Do you have any intuition or insights you can share with us as to why this would be the case? Our experiments with tit-for-tat strategies demonstrated that, in the more complex setting, achieving both literal theory of mind and functional theory of mind is more difficult. This is inline with our general expectation. We also would maintain that this problem is still concerning even if it were the case that it appeared to go away (for some reason) in more complex scenarios. To make an analogy with mathematical reasoning, even if a model performed well on the competitive AIME benchmark, we believe it would still be very troubling if it could not correctly answer $1+1=2$. We view our Rock, Paper, Scissors against a fixed agent scenario as akin to the $1+1=2$ for theory of mind. In fact, in our initial experiments, we tried much more complex scenarios and iteratively simplified the setting as we continued to see poor performance.
>
> **Concern “relatively short horizon”:** We should point out that our experiments considered a horizon of 100 steps, which was 10 times longer than prior work exploring these domains with LLMs. If you look through the plots we provided in Figures 19-43 in the appendix, you can see that in many cases there seems to be no evidence that performance will continue to get better or converge with more interactions. We actually believe our experiments are of a long horizon relative to the information content in the game and the partner policies, which is why we see that performance is already “asymptoting” in many cases. We are open to running longer experiments if this is still a point of concern for you.
>
> **Questions 1 and 2:** The primary scope of our paper is to establish that existing benchmarks can be misleading. As such, we believe actually proposing a new benchmark would not be appropriate for a position paper and should be left to future work. Our main insight is that benchmarks need an interactive component where the theory of mind reasoning performance of LLMs can be assessed separately from predictions about literal theory of mind. We will provide a section to clarify this at the end of our revised paper. As pointed out by reviewer Pqm1, Codenames is a good game for measuring theory of mind, we will also highlight Hanabi, Taboo, and Wavelength.
>
> **Question 3:** We also feel that the failure of social prompting strategies is a very surprising result in our paper. We actually believed it would work before we tried it. It clearly seems to be indicative of a failure for the LLMs to effectively reason over its inputs. Our hypothesis is that it is somewhat a kin to the needle in the haystack benchmarks considered for LLM long-context reasoning. Intuitively, we are kind of surprised it is still such a problem if this information is provided very close to the end of the prompt, but this indeed appears to be the case. As you can see in Table 3, it seems that the interaction history and payoff information tends to be very distracting to the agent in many cases.

---

### Official Review · Reviewer_sQuU · 2025-03-20

**Significance:** 3
**Argument Clarity:** 3
**Rating:** 4
**Confidence:** 3

**Questions:**

- The two definitions are not consistent. One defines a loss function and the other one defines a regret. Why the discrepancy?

**Discussion Potential:**

4

**Paper Summary:**

The paper argues that current theory of mind benchmarks for large language models (LLMs) are insuffiicent because they primarily measure literal theory of mind, which is the ability to predict behavior, and fail to assess functional theory of mind, the crucial ability to adapt behavior in-context based on those predictions. The authors contend that LLMs can achieve high scores on literal theory of mind benchmarks without demonstrating consistent reasoning or the capacity to use their predictions to inform their own actions effectively. Through experiments in simple matrix games, the paper shows that even top-performing LLMs struggle with functional theory of mind despite strong literal theory of mind performance. The authors conclude that new, more challenging benchmarks are needed to directly evaluate functional theory of mind in LLMs to ensure reliable performance in multi-agent social contexts

## Update after rebuttal
I thank the authors on the response and the intent to differentiate the theory of mind term. I will maintain my score.

**Position:**

Yes

**Position In Title:**

Yes

**Related Work:**

3

**Strengths And Weaknesses:**

Strengths:
* Introduction of a valuable conceptual framework: The paper introduces the terms "literal theory of mind" (Definition 2.1) and "functional theory of mind" (Definition 2.2) as distinct concepts, providing a useful framework for analyzing and discussing theory of mind in AI agents. These definitions help clarify what aspects of social intelligence are currently being measured and what is missing.

* Empirical demonstration of the gap: The paper presents experimental results using matrix games that demonstrate a significant gap between the literal and functional theory of mind performances.

* Exploration of various prompting strategies: The paper rigorously tests a wide range of prompting strategies, highlighting the challenges in eliciting functional theory of mind from LLMs through prompting alone.

* Analysis of the role of long contexts: The paper investigates whether the difficulty in achieving functional theory of mind stems from issues with reasoning over long interaction histories, finding that even with oracle access to the partner's actions, LLMs struggle to reason effectively.

* Consideration of inductive bias: The paper examines the impact of different action representations (neutral, canonical, repeated neutral, nonsense) on LLM performance, revealing that inductive bias can have both short-term benefits and long-term drawbacks for convergence.

Weaknesses:

* Theory of mind has a different connotation in multi-agent literature than in the psychology where it origins from. Historically the ambiguity hasn't been an issue because the fields are different. However, in the context on the LLMs, it is not immediately clear to which theory of mind the authors are referring. It does become clear later in the paper. I would suggest that the author frame the paper more specifically and disambiguate the terminology earlier in the manuscript.

**Support:**

3

---

> ### Author Rebuttal · Authors · 2025-04-01
>
> Thank you for your thoughtful review of our paper. We really appreciate your kind words regarding the value of the conceptual framework we propose and the rigor of our empirical analysis.  We have attempted to address your primary question and concern below.
>
> **Question - “definitions are not consistent”:** We tend to disagree and feel that the definitions are actually consistent, but we would be happy to follow up on this further during the discussion period if we did not correctly understand your point. Regret functions very similarly to a loss function. They both are metrics that we seek to minimize with zero as the best value. Both are also distances with respect to an optimal reference i.e. the reward received by the optimal policy for functional theory of mind, and the actions taken by the true policy of the other agent for literal theory of mind. One clear discrepancy in our notation is the use of $L$ in one case and $\Delta$. We think it would be reasonable to replace $\Delta$ with $L$, but made this particular choice to stay consistent with the common notation for each in the literature. However, we definitely see why this is confusing and makes it seem like the definitions are inconsistent. We will add a footnote to make this clear in the final draft of the paper.
>
> **Weakness - “not immediately clear to which theory of mind the authors are referring”:** Thank you for making this point. This is very useful feedback for us. In our updates to the paper, we will be sure to make it clear in the introduction that we are exploring theory of mind as it applies to reasoning in the behavioral multi-agent sense. It is definitely very important, as you mention, to also point out subject areas like psychology and philosophy where the term is used differently.

---

### Decision · Program_Chairs · 2025-04-30

**Decision:**

Accept (poster)

**Comment:**

This paper argues that current theory of mind (ToM) benchmarks for large language models (LLMs) are inadequate because they evaluate only literal ToM—predicting others’ actions—while failing to assess functional ToM—adapting behavior based on those predictions. The authors introduce formal definitions to distinguish these two capabilities and conduct experiments showing that LLMs can perform well on literal ToM tasks yet struggle with functional ToM, even in simple multi-agent settings.

The reviewers broadly agreed that the paper articulates a meaningful position and makes a useful conceptual contribution. Reviewer sQuU highlighted, "The paper introduces the terms 'literal theory of mind' (Definition 2.1) and 'functional theory of mind' (Definition 2.2) as distinct concepts, providing a useful framework for analyzing and discussing theory of mind in AI agents.” Reviewer sm5Y supported the distinction but questioned the generality of the experiments; the authors clarified in rebuttal that they included longer-horizon interactions and dynamic partners (e.g., tit-for-tat), with details in the appendix. Reviewer Pqm1 initially raised concerns that the formalism might “formulate any RL algorithm as a ToM task,” but after the authors clarified their intent, noted that “the paper is really interesting and has potential to foster discussion.” Reviewer 1Qq8 suggested linking the distinction between literal and functional ToM to the predict-then-optimize literature, which the authors welcomed as a valuable addition.